# Towards accurate differential diagnosis with large language models

Daniel McDuff[1,10]✉, Mike Schaekermann[2,10]✉, Tao Tu[3,10], Anil Palepu[4,10], Amy Wang[2], Jake Garrison[1], Karan Singhal[3], Yash Sharma[4], Shekoofeh Azizi[5], Kavita Kulkarni[4], Le Hou[4], Yong Cheng[6], Yun Liu[4], S. Sara Mahdavi[5], Sushant Prakash[3], Anupam Pathak[4], Christopher Semturs[4], Shwetak Patel[1], Dale R. Webster[4], Ewa Dominowska[1], Juraj Gottweis[7], Joelle Barral[8], Katherine Chou[4], Greg S. Corrado[4], Yossi Matias[4], Jake Sunshine[1,11]✉, Alan Karthikesalingam[9,11]✉ & Vivek Natarajan[4,11]✉

A comprehensive differential diagnosis is a cornerstone of medical care that is often reached through an iterative process of interpretation that combines clinical history, physical examination, investigations and procedures. Interactive interfaces powered by large language models present new opportunities to assist and automate aspects of this process[1]. Here we introduce the Articulate Medical Intelligence Explorer (AMIE), a large language model that is optimized for diagnostic reasoning, and evaluate its ability to generate a differential diagnosis alone or as an aid to clinicians. Twenty clinicians evaluated 302 challenging, real-world medical cases sourced from published case reports. Each case report was read by two clinicians, who were randomized to one of two assistive conditions: assistance from search engines and standard medical resources; or assistance from AMIE in addition to these tools. All clinicians provided a baseline, unassisted differential diagnosis prior to using the respective assistive tools. AMIE exhibited standalone performance that exceeded that of unassisted clinicians (top-10 accuracy 59.1% versus 33.6%, $P = 0.04$). Comparing the two assisted study arms, the differential diagnosis quality score was higher for clinicians assisted by AMIE (top-10 accuracy 51.7%) compared with clinicians without its assistance (36.1%; McNemar's test: 45.7, $P < 0.01$) and clinicians with search (44.4%; McNemar's test: 4.75, $P = 0.03$). Further, clinicians assisted by AMIE arrived at more comprehensive differential lists than those without assistance from AMIE. Our study suggests that AMIE has potential to improve clinicians' diagnostic reasoning and accuracy in challenging cases, meriting further real-world evaluation for its ability to empower physicians and widen patients' access to specialist-level expertise.

An accurate diagnosis is a critical component of effective medical care. Building artificial intelligence (AI) systems that are capable of performing or assisting clinicians in this important task has been a long-standing grand challenge[2]. Whereas prior focus has been on evaluating a machine's ability to accurately output a diagnosis[1,3–5], real-world clinical practice involves an iterative and interactive process of reasoning about a differential diagnosis (DDx), weighing multiple diagnostic possibilities in the light of increasing amounts of clinical information over time. Deep learning has been applied to promising effect for generating DDx in a number of specialties including radiology[4], ophthalmology[5] and dermatology[3], but such systems lack the interactive capabilities to fluently assist a user through communication in natural language.

The emergence of large language models (LLMs) presents an opportunity to design novel interactive tools and interfaces to aid DDx. These models have demonstrated the ability to perform complex language comprehension and reasoning tasks, generating coherent text and thereby enabling a large variety of real-world applications[6–9]. Both general-purpose LLMs (GPT-4) and medical domain-specialized LLMs (Med-PaLM 2) have demonstrated strong performance in standardized and multiple-choice medical benchmarks[10,11]. Such evaluations represent a natural starting point for probing the model's medical knowledge and capabilities but do not measure utility in real-world scenarios for care delivery—for example, in challenging medical cases faced by trained physicians. It is also not obvious how these models might actively assist clinicians in the development of a DDx. Recent work has begun to assess the standalone performance of these models on challenging case reports that involve complex deduction and diagnosis[1,12–14], but has stopped short of evaluating how they can assist clinicians, augment performance and empower them to provide better care.

[1]Google Research, Seattle, WA, USA. [2]Google Research, Toronto, Ontario, Canada. [3]Google Research, New York City, NY, USA. [4]Google Research, Mountain View, CA, USA. [5]Google DeepMind, Toronto, Ontario, Canada. [6]Google DeepMind, Mountain View, CA, USA. [7]Google Research, Zurich, Switzerland. [8]Google DeepMind, Paris, France. [9]Google Research, London, UK. [10]These authors contributed equally: Daniel McDuff, Mike Schaekermann, Tao Tu, Anil Palepu. [11]These authors jointly supervised this work: Jake Sunshine, Alan Karthikesalingam and Vivek Natarajan. ✉e-mail: dmcduff@google.com; mikeshake@google.com; jakesunshine@google.com; alankarthi@google.com; natviv@google.com

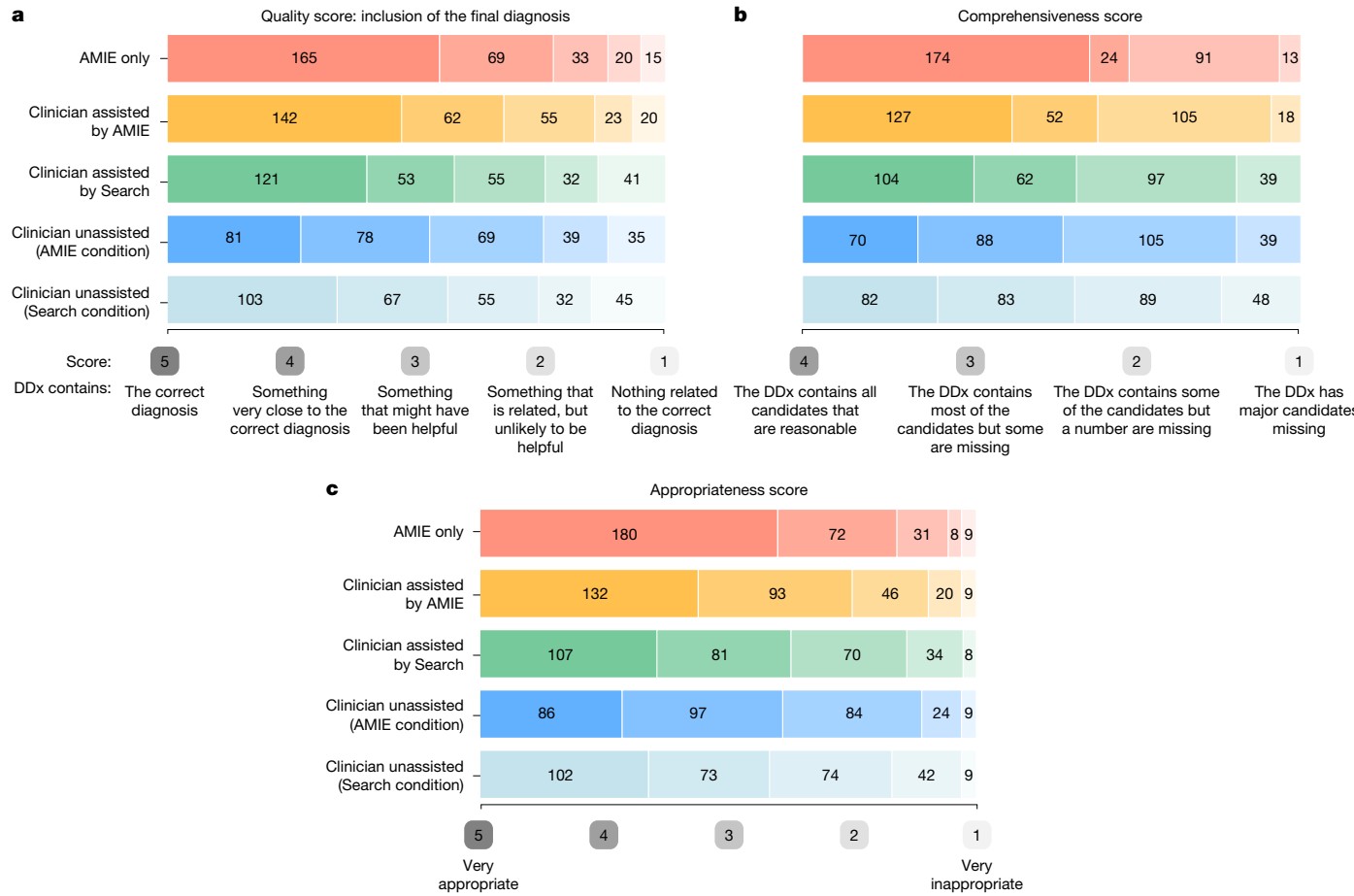

**Fig. 1 | Evaluation of the quality of DDx lists from generalist physicians. a**, DDx quality score based on the question: "How close did the differential diagnoses (DDx) come to including the final diagnosis?" **b**, DDx comprehensiveness score based on the question: "Using your DDx list as a benchmark/gold standard, how comprehensive are the differential lists from each of the experts'?" **c**, DDx appropriateness score based on the question: "How appropriate was each of the DDx lists from the different medical experts compared to the differential list that you just produced?" The colours correspond to experiment arms, and the shade of the colour corresponds to different levels on the rating scales. In all cases, AMIE and clinicians assisted by AMIE scored highest overall. Numbers reflect the number of cases (out of 302). Note that the clinicians had the option of answering "I am not sure" in response to these questions; they used this option in a very small number (less than 1%) of cases.

Here we introduce AMIE, an LLM that is optimized for clinical diagnostic reasoning to generate a DDx for challenging, real-world medical cases. Beyond measuring standalone performance, we integrated this model into an interactive interface to measure how well AMIE could assist clinicians in developing a DDx. Using a set of challenging real-world case reports from the *New England Journal of Medicine* (NEJM) clinicopathological conferences (CPCs), we compared clinicians' ability to form a DDx with the assistance of AMIE versus with access to traditional information retrieval tools (such as internet searches and books). AMIE achieved impressive performance in both generating DDx lists that contained the correct diagnosis (top-10 accuracy) and in identifying the correct final diagnosis as the most likely in the list (top-1 accuracy). Under automated model-based evaluation, the quality and the accuracy of the DDx list produced by AMIE was found to be significantly better than the state-of-the-art GPT-4 model available at the time of the experiments[1]. Perhaps more importantly, AMIE also improved the diagnostic capability of clinicians as measured by the quality of their DDx lists for the evaluated cases. LLMs optimized for the safety-critical medical domain such as ours present a novel paradigm for assisting clinicians because of the potential for variation in the ways in which a given individual may converse with the system and utilize it in collaborative reasoning.

A detailed explanation of the cases, their components, how they were fed to the model, the randomization scheme of AMIE versus the standard practice and information on the expert raters of the model

and how the outputs were evaluated by blind expert raters, can be found in Methods.

In evaluating the quality of the DDx lists we used several criteria, inspired by the approach taken in ref. 1 and extended to draw additional insight from the clinicians. First, we measured whether the final diagnosis matched an entry in the DDx list and in which position (top-*n* accuracy). Second, we used the quality score from Bond et al.[15] and created appropriateness and comprehensiveness scales. Combined, these measures assess overall DDx quality, appropriateness and comprehensiveness.

When using AMIE for assistance, clinicians asked, on average (mean), 2.92 questions in the interface (median 2, interquartile range (IQR) 1–4). On average (mean), clinician questions consisted of 9.39 words (median 10, IQR 6–12) and 54.31 characters (median 61, IQR 39-63). AMIE's responses, on average (mean), consisted of 237.60 words (median 198, IQR 127–332) and 1,540.81 characters (median 1,276; IQR 815–2210).

In the Search condition, the most popular tools were UpToDate (used in 34% of tasks), Google Search (30%) and PubMed (22%). Although clinicians were allowed to use additional tools in the AMIE condition, this was far less frequent (less than 5% of tasks).

## DDx performance of AMIE

The DDx lists produced by our language model achieved strong quality, appropriateness and comprehensiveness scores (see Fig. 1). The median

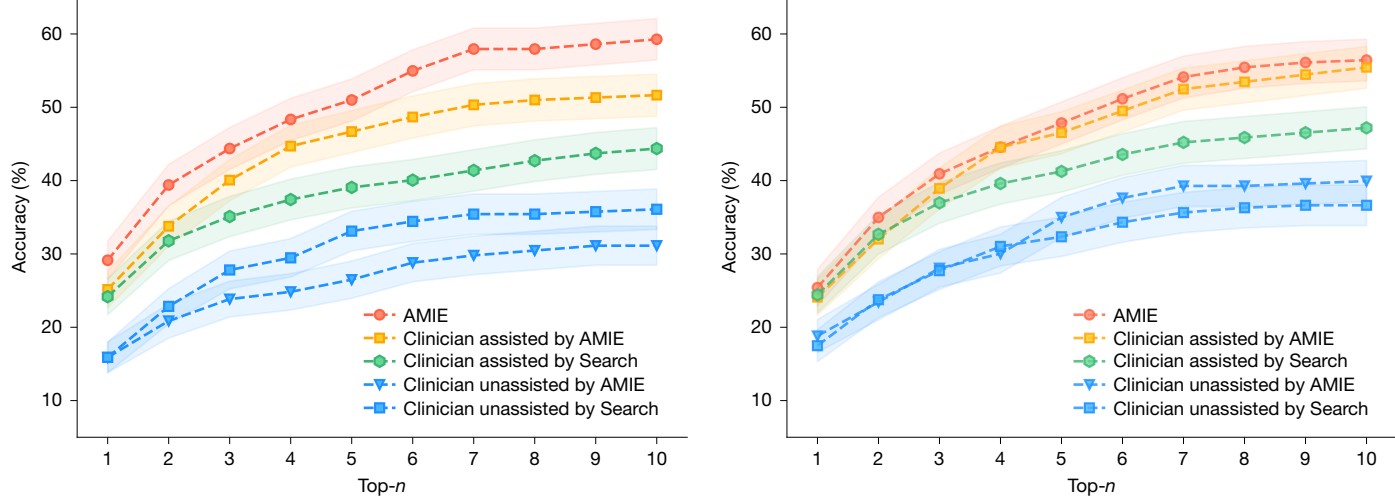

**Fig. 2 | Top-*n* accuracy in DDx lists through human and automated evaluations.** The percentage accuracy of DDx lists with the final diagnosis through human evaluation (left) or automated evaluation (right). Points reflect the mean; shaded areas show ±1 s.d. from the mean across 10 trials.

quality score was 5 ('DDx includes the correct diagnosis') with 54% of DDx lists achieving that score. The number of cases that scored 5 (that is, the DDx included the top diagnosis) was statistically significantly higher for AMIE compared with clinicians without assistance (McNemar's test: 64.4, $P < 0.01$). The mean appropriateness score was 4.43 out of 5 (s.d. 0.92). The median comprehensiveness score was 4 ('The DDx contains all candidates that are reasonable') with 55% of the DDx lists achieving that score.

The mean appropriateness score of AMIE (4.34) was significantly higher than that of unassisted clinicians (3.74) (paired *t*-test 8.52, $P < 0.001$, Wilcoxon signed-rank test: 2,857.5, $P < 0.001$) and assisted clinicians in either the Search (3.80) (paired *t*-test 7.23, $P < 0.001$, Wilcoxon signed-rank test: 3,308.5, $P < 0.001$) or LLM (4.06) (paired *t*-test 4.98, $P < 0.001$, $P < 0.001$, Wilcoxon signed-rank test: 2,752.0, $P < 0.001$) conditions.

For computing top-*n* accuracy, if any of the first *n* diagnoses in an individual DDx were marked correct by the language model, the differential was considered to be correct. We computed the proportion of correct DDx lists across all cases to compute the top-*n* accuracy (for *n* from 1 to 10) for each DDx. AMIE reliably generated DDx lists that perform well against the ground truth diagnosis (Fig. 2). AMIE provided the correct diagnosis in 177 (59%) of the DDx lists and in 89 (29%) of the lists it was at the top of the list. These scores are higher than those achieved by the clinicians in any of the conditions. The top-10 accuracy of AMIE (59.1%) was significantly higher than the top-10 accuracy for the unassisted clinicians (33.6%) ($P = 0.04$) (Tables 1 and 2).

Figure 2 shows the top-*n* accuracy based on human and the automated metric. The results are broadly similar, illustrating that despite the final diagnoses often being complex and nuanced, the automated metric faithfully captures the distinction between a DDx list that includes the correct diagnosis and one that does not.

The clinicians in the study were not required to give a full list of ten diagnoses for every case. Clinicians in conditions I and II were required to given a minimum of three diagnoses. The median number provided was six. The performance at $n = 6$ is of particular relevance. Not all clinicians provided six diagnoses, as a result we conducted a variable top-*n* experiment, where for each case *n* was set to the number of diagnoses provided by the human clinicians. The variable top-*n* performance of AMIE was 59.4%—this is similar to the performance at $n = 9$ and at $n = 10$. As a result, AMIE's output of a full list of ten diagnoses did not place it at an advantage compared to the clinicians.

## AMIE as a DDx assistant

Of the DDx lists created before assistance 37% (Search condition) and 29% (AMIE condition) achieved a quality score of 5 (Fig. 1). For comparison, 49% of those created with assistance from AMIE scored 5.

The number of cases that scored 5 (that is, the DDx included the top diagnosis) was statistically higher for clinicians assisted by AMIE compared with clinicians without assistance (McNemar's test: 48.3, $P < 0.01$) and clinicians with Search assistance (5.45, $P = 0.02$).

For comprehensiveness, the number of cases that scored 4 (that is, The DDx contains all candidates that are reasonable) was statistically higher for clinicians assisted by AMIE compared with clinicians without assistance (McNemar's test: 185.8, $P < 0.01$) and clinicians with Search assistance (185.8, $P < 0.01$). As a consistency check, the number of cases that scored 4 was not statistically higher for clinicians in the Search condition (I) baseline and AMIE condition (II) baseline (McNemar's test: 1.47, $P = 0.23$).

The mean appropriateness score after assistance with AMIE (4.06) was significantly higher than after assistance with Search (3.80) (paired *t*-test 3.32, $P = 0.001$) and the baseline (3.74) (paired *t*-test 4.79, $P < 0.001$).

To summarize, with the support of AMIE, the quality, appropriateness and comprehensiveness scores for the DDx lists were greater than for the lists prior to assistance (see Fig. 1).

The top-*n* accuracy of the clinicians increased with assistance from AMIE compared to without (see Fig. 2). A Sankey diagram illustrates the effect of the two forms of assistance (Search and AMIE) on top-10 accuracy (Fig. 3). In the AMIE condition, 73 cases that did not feature the final diagnosis prior to using the tool included it after assistance from AMIE. This result is in contrast to only 37 cases in the Search condition. Comparing the two assisted study arms, the DDx quality score was higher for clinicians assisted by AMIE (top-10 accuracy 51.7%) compared with clinicians without its assistance (36.1%) (McNemar's test: 45.7, $P < 0.01$) and clinicians with search (44.4%) (4.75, $P = 0.03$).

## Task duration with AMIE and Search

The time taken to generate updated DDx lists in the Search condition versus the AMIE condition were similar (Search: 7.19 ± 5.33 min, AMIE: 7.29 ± 6.41 min (mean ± s.d.)). These were not significantly different (paired *t*-test $P = 0.807$), which is surprising as the clinicians all had experience using internet search and other information retrieval

**Table 1 | Top-1 and top-10 accuracy of DDx lists produced with AMIE and Search assistance**

| | Model only | | Human | | | | | |
|---|---|---|---|---|---|---|---|---|
| | AMIE | | Before assistance | | After Search assistance | | After AMIE assistance | |
| Metrics | Top-1[↑] | Top-10[↑] | Top-1[↑] | Top-10[↑] | Top-1[↑] | Top-10[↑] | Top-1[↑] | Top-10[↑] |
| Full set (302 cases) | 29.2% | 59.1% | 15.9% | 33.6% | 24.3% | 44.5% | 25.2% | 51.8% |
| Set with no overlap (56 cases) | 35.4% | 55.4% | 13.8% | 34.6% | 29.2% | 46.2% | 24.6% | 52.3% |
| Difference compared to full set | **+6.2%** | **−3.7%** | **−2.1%** | **+1.0%** | **+4.9%** | **+1.7%** | **−0.6%** | **+0.5%** |
| Set with partial overlap (249 cases) | 29.9% | 61.4% | 14.9% | 33.1% | 24.3% | 44.2% | 24.7% | 51.4% |
| Difference compared to full set | **+0.7%** | **+2.3%** | **−1.0%** | **−0.5%** | **0%** | **−0.3%** | **−0.5%** | **−0.4%** |

The percentage of DDx lists with the final diagnosis. Bold numbers reflect the difference in percentage accuracy between the full case set and the partial case sets.

tools, yet they were using the AMIE interface for the first time. We had hypothesized that they would take longer using AMIE owing to the initial learning curve.

## Length of DDx lists with AMIE and Search

When unassisted, the median length of the DDx lists was 6 (IQR 5–9); the mean was 6.41 (s.d. 2.39). With search the median DDx list length was 7 (IQR 5–10); the mean was 6.92 (s.d. 2.52). With AMIE, the median DDx list length was 8 (IQR 6–10); the mean was 7.58 (s.d. 2.33). With assistance from AMIE, the length of the DDx lists was longer than without assistance (paired $t$-test: 7.13, $P < 0.001$) and longer than the DDx lists with assistance from search (paired $t$-test: 3.15, $P = 0.002$).

## AMIE comparison with GPT-4

As we did not have the same set of human raters who evaluated the differentials produced by GPT-4[1] and AMIE, we cannot compare top-10 accuracy numbers directly. Therefore, in our study design, we evaluate performance on that 70-case subset (reported in ref. 1) using the automated metric (which is shown above to be relatively consistent with human evaluation). AMIE performs better with regard to top-$n$ accuracy for $n > 1$, with the gap being most prominent for $n > 2$ (Fig. 4). This suggests potentially significant improvements in quality and comprehensiveness of the differentials produced by AMIE. For $n = 1$, GPT-4 performs marginally better but not statistically significantly.

## Discussion

We used a popular series of complex diagnostic challenges to evaluate an LLM optimized for clinical reasoning and diagnosis (AMIE); both in a standalone capacity and under randomized comparisons as an assistive tool for physicians. In standalone performance, AMIE generated more appropriate and comprehensive DDx lists than physicians when they were unassisted, with its DDx lists being more likely to include the final diagnosis than DDx lists from a board-certified internal medicine physician, regardless of what position in the DDx list was considered (that is, top-$n$ accuracy with $n$ ranging from 1 to 10). Clinicians using AMIE as an assistant produced a DDx with higher top-$n$ accuracy, and DDx with greater quality, appropriateness and comprehensiveness compared with the status quo for clinical practice (use of internet search and other resources).

The NEJM CPCs examined here are well-known for being unique and challenging clinical conundrums. Within this distinctive setting, AMIE outperformed an unassisted board-certified physician in both top-1 and top-$n$ accuracy. Whereas the CPCs have long been used as benchmarks for difficult diagnosis, it is also well-known that performance in CPCs in no way reflects a broader measure of competence in a physician's duties[16]. Furthermore, the act of forming a DDx comprises many other steps that are not scrutinized in this study, including the goal-directed acquisition of information under uncertainty (which is known to be challenging for AI systems despite recent technical progress in this direction[17–19]).

We are therefore very cautious in extrapolating our findings towards any implications about the utility of AMIE as a standalone diagnostic tool. Nevertheless, our controlled evaluation mirrored the findings of other recent works exploring the performance of LLMs and pre-LLM 'DDx generators' in smaller subsets of the NEJM CPCs, which have shown the potential for automated technology to reach the correct DDx with superior performance to standalone physicians in these challenging cases[1,12,13,20]. Although this represents a step beyond historical attempts at automating DDx in NEJM CPCs, in which computerized approaches were deemed overtly unreliable for practical use[21], such studies also undertook limited consideration of the quality of DDx generated by these automated systems or their role as assistive tools.

Our work extends previous observations by showing not only that AMIE was more likely to arrive at a correct answer or provide the correct answer in a list, but also that its DDx were determined by an independent rater to be of higher appropriateness and comprehensiveness than those produced by board-certified physicians with access to references and search.

In our study, clinicians had access to both images and tabular data in redacted case reports, whereas AMIE was only provided with the main body of the text. Although AMIE outperformed the clinicians despite this limitation, it is unknown whether and how much this gap would widen if AMIE had access to the figures and tables. Furthermore,

**Table 2 | Top-1 and top-10 accuracy of DDx lists produced with AMIE and Search assistance by speciality**

| | Model only | | Human | | | | | |
|---|---|---|---|---|---|---|---|---|
| | AMIE | | Before assistance | | After Search assistance | | After AMIE assistance | |
| Metrics | Top-1[↑] | Top-10[↑] | Top-1[↑] | Top-10[↑] | Top-1[↑] | Top-10[↑] | Top-1[↑] | Top-10[↑] |
| Internal medicine (159 cases) | 27.7% | 61.6% | 15.5% | 34.6% | 24.5% | 47.8% | 24.5% | 52.8% |
| Neurology (42 cases) | 26.8% | 56.1% | 17.1% | 31.7% | 22.0% | 36.6% | 24.4% | 51.2% |
| Paediatrics (33 cases) | 30.3% | 45.5% | 6.1% | 22.7% | 12.1% | 33.3% | 15.2% | 30.3% |
| Psychiatry (10 cases) | 50.0% | 70.0% | 20.0% | 50.0% | 20.0% | 60.0% | 30.0% | 60.0% |

The percentage of DDx lists with the final diagnosis by specialty.

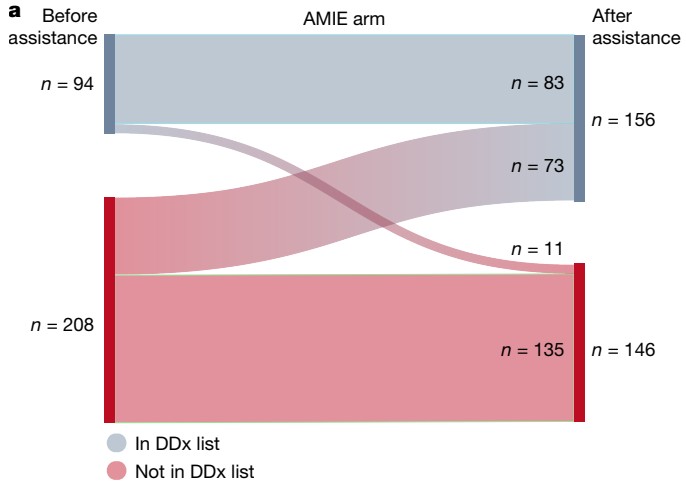

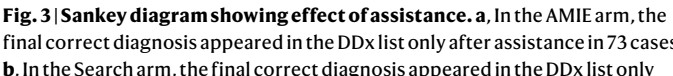

- In DDx list
- Not in DDx list

**Fig. 3 | Sankey diagram showing effect of assistance. a**, In the AMIE arm, the final correct diagnosis appeared in the DDx list only after assistance in 73 cases. **b**, In the Search arm, the final correct diagnosis appeared in the DDx list only after assistance in 37 cases. In a small minority of cases in both arms (AMIE arm: 11 (**a**); Search arm: 12 (**b**)), the final diagnosis appeared in the DDx list before assistance but was not in the list after assistance.

the integration of multimodal inputs by LLMs is an area of novel research[22,23], with a large potential number of data modalities to consider and little precedent for how information from multiple modalities should be integrated over time for a single case by AI systems.

The repeated examination of NEJM CPCs by automated systems highlights its promise as a 'benchmark' for evaluation and development of LLMs. Benchmarking enables comparisons of models with one another and the ability to evaluate a model's performance improvements or degradation over time. However, consistency in using CPCs as a scalable benchmark is challenging if we are reliant on using human judgement to establish whether a candidate DDx matches the ground truth. We utilized an automated approach for comparing AMIE to a baseline LLM performance (GPT-4). Our estimates varied from recently published estimates in other studies, despite using the same subset of cases[1]. Direct comparisons of different technologies would ideally be conducted by more extensive and blinded human evaluation, including work to ensure reproducibility of the human evaluation protocol, analysis of inter-rater reliability and the use of metrics that reflect the quality, appropriateness and comprehensiveness of LLM differentials in addition to estimations of accuracy. Our estimates of top-1 and top-10 accuracy, although impressive at close to 30% and 60%, respectively, highlight noteworthy room for improvement for LLMs, especially for complex cases that are non-pathognomonic (that is, cases that do not have a sign or symptom that defines a diagnosis). However, as noted above, the CPCs represent 'diagnostic puzzles' rather than real-world examples of common clinical workflows, and it is therefore important to consider more realistic settings in which LLMs might prove of practical value in medicine.

One such example is the potential for LLMs to assist clinicians in complex diagnoses. Deep learning tools have shown considerable promise in many areas of medicine, but are overwhelmingly used as assistive rather than autonomous tools[24], given the safety-critical nature of medical practice and the many issues of robustness[25] and fairness[26–28] seen in deployment. Furthermore, observations of standalone diagnostic accuracy often do not guarantee that an AI tool will improve performance in real-world settings as an assistive tool, and it remains unclear how AI and human decision-making should be optimally integrated in medicine[29]. For LLMs in particular, the known incidence of hallucination and confabulation[30] might mislead clinicians into inaccurate diagnosis, replicating or even extending findings in other clinical settings that AI systems might actually degrade the performance of clinicians rather than necessarily improving outcomes.

This highlights the importance of focused study of LLMs in assistive scenarios. We explored this specifically in NEJM CPCs and found that AMIE increased the number of appropriate DDx produced by a clinician when used as an assistive tool in addition to overall top-*n* accuracy, suggesting that AMIE's primary assistive potential may be due to making the scope of DDx more complete. Given the potential for misleading information to arise from AI systems, including in convincing dialogue, clinicians must appreciate the fundamental limitations of these models and not lose sight of their primacy in the provider–patient relationship and their ultimate authority and responsibility for the diagnostic and therapeutic management of their patients. Such thoughtful and effective LLM use should not be unintuitive to most clinicians. Aiding the diagnostic process could reasonably occur in an emergency room upon presentation (during potentially time-sensitive moments), upon admission to the medical ward, or by a consulting service after a patient has been admitted or in outpatient clinics. Our findings suggest that future research should more rigorously explore how LLMs augment clinicians' DDx in many such specific scenarios, where the risks and benefits might vary.

Despite being a novel tool, the use of AMIE did not seem to add inefficiency or increase the amount of time spent on solving each CPC compared with the use of Search or other conventional information. This suggests that the conversational interface was unobtrusive and intuitive. Consistent with this, the interviewed clinicians all described it as 'easy' to use, and were positive about the use and implications of the AMIE interface. Enhancing efficiency while maintaining or improving quality are generally accepted goals of improving healthcare delivery, alongside improving provider experience[31], and our study showed significant potential in this regard, as clinicians also reported feeling more confident in their DDx lists after using the model. The clinicians described search becoming difficult when they did not know how to start or narrow down the query; qualitatively, the reports indicate that AMIE was easier to use in this regard. However, there are many human factors, social elements and other complex considerations in these use cases, and it is critical to ensure that efforts are made to avoid inequities in access to avoid exacerbating existing health disparities.

Clinicians frequently expressed excitement about using AMIE, but were also aware of the shortcomings of language models and had concerns about confabulations in particular if used by individuals who were not trained or instructed to avoid such questions. However, our work did not explore many other important aspects of human–AI interaction, which require further study in safety-critical settings such as this. For example, we did not explore the extent to which clinicians

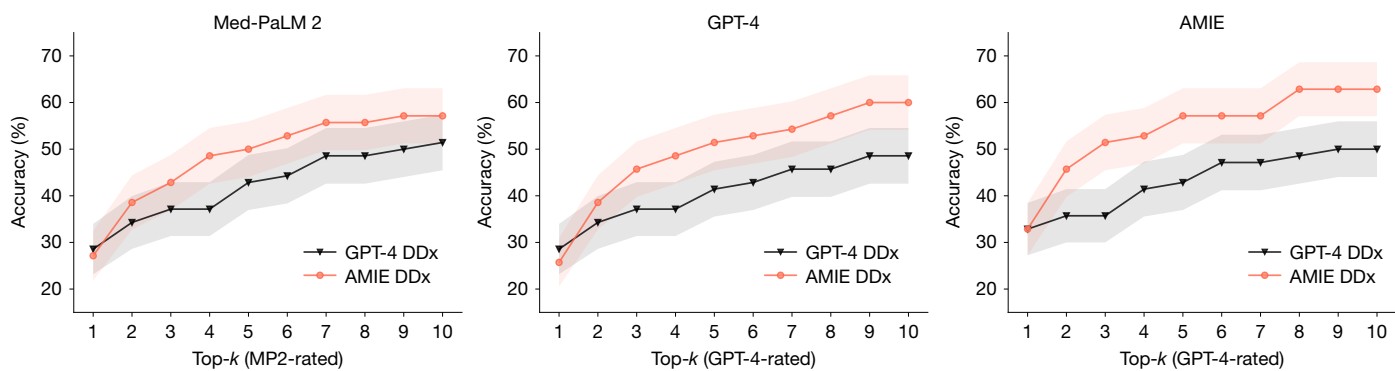

**Fig. 4 | Top-*n* accuracy in DDx lists from different LLMs.** Comparison of the percentage of DDx lists that included the final diagnosis for AMIE versus GPT-4 for 70 cases. We used Med-PaLM 2[10], GPT-4[6] and AMIE as the raters—all resulted in similar trends. Points reflect the mean; shaded areas show ±1 s.d. from the mean across 10 trials.

trusted the outputs of the model or their understanding of its training and limitations, or undertake focused 'onboarding' or training in its use, which are all known to be important modifiers of the benefits derived by clinicians from AI assistants[32]. The CPC challenges themselves do not enable a rigorous exploration of the possible effects of AI assistance on health equity and fairness; a further study of how these aspects of clinicians' DDx is affected by LLM assistance is needed. AI systems are known to be able to express uncertainty[33] and defer appropriately to clinicians[34], which might significantly improve the balance between trust and skepticism needed for effective AI assistance in medicine. Qualitative feedback suggested that there remains room for targeted improvement of LLMs as assistive diagnostic tools, with one clinician noting that "It was most helpful for simpler cases that were specific keywords or pathognomonic signs", but for more complex cases it still tended to draw conclusions from isolated symptoms rather than viewing the case holistically. The assistive effect of these LLMs could potentially 'upskill' clinical providers, particularly in enabling them to broaden and enhance the quality of their DDx. As corroborated via our clinician interviews after their experience with AMIE, such upskilling could be relevant for education or training purposes to support providers across a skill continuum ranging from trainees to attending providers. The upskilling capabilities could also extend to locations where specialist medical training is less common (such as in lower and middle income countries). However, our findings may not generalize to these scenarios, given that we utilized a pool of 20 clinicians with a mean experience of 11.5 years. This may not adequately represent the diverse set of users who are seeking to benefit from LLMs as a diagnostic aid.

Our qualitative findings from semi-structured interviews with clinicians highlight the collaborative nature of the diagnostic reasoning process and the importance of clinical judgement when using an LLM. Whereas AMIE was capable of generating a broad DDx in isolation, the clinicians' expertise enabled them to filter these suggestions when they were using the tool, discarding those they deemed to be inaccurate or irrelevant and leading to a more comprehensive and considered final differential list. This active evaluation and filtering process could explain the gap between standalone AMIE performance and clinician performance when assisted by the tool, with several specific factors highlighted: (1) anchoring bias: clinicians tended to anchor on their initial, unassisted DDx. This is consistent with known anchoring biases and might be exacerbated by the two-stage study design; (2) LLM suggestibility: several clinicians noted that AMIE could be led down alternative diagnostic paths by their follow-up questions and that this could lead to inaccurate conclusions that clinicians recognized as not being supported by the evidence; (3) trust calibration: clinicians highlighted the importance of the model being able to communicate when it is unsure, as this would probably have influenced the extent to which they trusted and incorporated AMIE's suggestions.

## Limitations

The NEJM CPC format differs in important ways from how a clinician would evaluate a patient at the outset of a clinical encounter. The case reports are created as 'puzzles' with enough clues that should enable a specialist to reason towards the final diagnosis. At the beginning of a clinician encounter, it would be challenging to create such a concise, complete and coherent case report. Case reports in the NEJM style would not be available at patient intake. Similarly, these cases were selected to represent challenging cases instead of common conditions. Thus, our evaluation does not directly suggest that clinicians should leverage the assistive capabilities of an LLM for typical cases that are seen on a daily basis.

Evaluation is non-trivial for complex tasks such as these case studies. Although the rubric that we used for evaluating whether a diagnosis is included in a DDx list is clear, it is possible to disagree whether an individual diagnosis is specific enough to be counted as correct versus incorrect. This ambiguity is likely to be the reason that we did not obtain identical results to Kanjee et al.[1].

In terms of modalities, the case reports include both images and tables. The clinicians had access to these in the redacted case reports. However, AMIE only had access to the main body of the text. Although AMIE outperformed the clinicians despite this limitation, it is unknown whether and how much this gap would widen if AMIE had access to the figures and tables. Early evidence suggests that the effect might be case and context dependent[13]. New multimodal models should be evaluated in a similar manner. The appropriate input format for images is clear, whereas tables can be represented textually or graphically. Experimentation into the optimal format for tabular data would also be valuable.

The study highlighted some weaknesses of AMIE. Specifically, one clinician (C3) highlighted that "It was most helpful for simpler cases that were specific keywords or pathognomonic signs" and that for more complex cases it still tended to draw conclusions from isolated symptoms rather than viewing the case holistically. Considering the importance of assessing challenging cases, the NEJM CPC case reports are likely to serve as a useful dataset for continued LLM benchmarking.

Regarding the time taken, we acknowledge that the analysis of time spent on the tasks may not map well to how an LLM would affect time on task in reality. We appreciate that in practice a clinician would need to write a case description or notes before being able to leverage this type of system.

There were potentially systematic differences between the clinicians' and the model's DDx lists that could have led the clinicians to guess that the lists came from different sources. However, we believe that this did not affect our results for several reasons. First, we reviewed the lists from the model and the clinicians before running the evaluation to ensure that there were no obvious formatting differences. Second, the raters did not know ahead of time the various potential sources of

DDx lists and that these could include AI models, and DDx list ordering was blinded during the rating process. Third, auto evaluation is blinded to the source of the data, and the trends from human and auto evaluation were consistent.

## Conclusion

Generating a DDx is a critical step in clinical case management, and the capabilities of LLMs present new opportunities for assistive tooling to help with this task. Tables 1 and 2 Our randomized study showed that AMIE was a helpful AI tool for DDx generation for generalist clinicians. Clinician participants indicated its utility for learning and education, and additional work is needed to understand its suitability for clinical settings.

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

## Methods

### NEJM CPC case reports

The case records of the Massachusetts General Hospital (MGH) are published, lightly edited transcriptions of the CPCs of the MGH (Boston, MA). In the CPC, a patient case presentation is described and then an expert physician is asked to provide a DDx and a final diagnosis, along with their diagnostic reasoning, based only on the patient's provided medical history and preliminary test results. The published cases, organized generally as diagnostic puzzles culminating in a definitive, pathology-confirmed diagnosis, are published regularly in the NEJM. We leverage these case reports, licensed from the NEJM, to evaluate AMIE's capability to generate a DDx alone and, separately, to aid clinicians in generation of their own differential. For this latter task, we developed a user interface for clinicians to interact with AMIE.

A set of 326 case texts from the NEJM CPC series were considered. These case reports were published over a 10-year period between 13 June 2013 and 10 August 2023. Of these, 23 (7%) were excluded on the grounds that they discussed case management and were not primarily focused on diagnosis. The articles were distributed over the years between 2013–2023 as follows—2013: $n$ = 22; 2014: $n$ = 34; 2015: $n$ = 36; 2016: $n$ = 35; 2017: $n$ = 36; 2018: $n$ = 16; 2020: $n$ = 23; 2021: $n$ = 36; 2022: $n$ = 39; 2023: $n$ = 26. Supplementary Table 2 contains the full list of case reports, including the title, year and issue number of each report. The 302 cases include the 70 cases used by Kanjee et al.[1].

These case reports cover a range of medical specialties. The largest proportion are from internal medicine ($n$ = 159), followed by neurology ($n$ = 42), paediatrics ($n$ = 33) and psychiatry ($n$ = 10). The text corresponding to the history of the present illness (HPI) was manually extracted from each article as input to AMIE. The average (median) word count of these sections of the case reports is 1,031 words (mean: 1,044, s.d.: 296, range: 378–2,428). The average (median) character count is 6,619 characters (mean: 6,760, s.d.: 1,983, range: 2,426–15,196).

A modified version of the article, inclusive of the provided HPI, admission imaging and admission labs (if available in the case) was created for the human clinicians (see Extended Data Fig. 1). This version had redacted the final diagnosis, expert discussion of the DDx and any subsequent imaging or biopsy results (which are typical elements of the conclusion of the case challenges). Given AMIE is a text-only AI model, the admission images and lab tables were not fed into the model. However, text-based descriptions of specific lab values or imaging findings were sometimes included in the case description.

### Training an LLM for DDx

Our study introduces AMIE, a model that uses a transformer architecture (PaLM 2[7]), fine-tuned on medical domain data; alongside an interface for enabling its use as an interactive assistant for clinicians.

As with Med-PaLM 2[10], AMIE builds on PaLM 2, an iteration of Google's LLM with substantial performance improvements on multiple LLM benchmark tasks. For the purposes of this analysis the large (L) PaLM 2 model was used.

AMIE was fine-tuned with long context length on a task mixture consisting of medical question answering (multiple-choice and long-form questions), medical dialogue generation and electronic health record (EHR) note summarization. The datasets used included the training splits of MultiMedQA (MedQA, MedMCQA, HealthSearchQA, LiveQA and MedicationQA)[10], a proprietary dataset of medical conversations, and expert handcrafted EHR note summaries from MIMIC-III[35]. The capability to process long context input enables AMIE to handle tasks that require long-range reasoning and comprehension.

From MedQA (multiple-choice) we used US Medical Licensing Examination (USMLE) multiple-choice style open domain questions with four or five possible answers[36]. A set of 11,450 questions were used for training and 1,273 questions reserved for testing. We curated 191

MedQA questions from the training set where clinical experts crafted step-by-step reasoning leading to the correct answer. From the Health-SearchQA, LiveQA and MedicationQA datasets we used expert-crafted long-form responses to 64 questions. The medical conversations were from a dataset of 218 dialogues between a clinician and patient that are associated with a corresponding ground truth diagnosis. The dataset features respiratory ($n$ = 168), musculoskeletal ($n$ = 40), cardiology ($n$ = 5), gastroenterology ($n$ = 4) and dermatology ($n$ = 1) dialogues. In total, 102 unique diagnoses are represented with the mean number of turns 98 (25th percentile = 85.0, 75th percentile = 113.0).

No NEJM case report data was used in the tuning process and we perform contamination experiments to assess whether performance was impacted by case text that may have been in the pretraining corpus.

**Zero-shot prompting.** We evaluated AMIE on each of the NEJM case studies with the following prompt: 'You are a helpful medical assistant. You will be provided and asked about a complicated clinical case; read it carefully and then provide a diverse and thorough DDx'.

**Example input.**
Case: [Case Text]
What are the top 10 most likely diagnoses? Be precise, listing one diagnosis per line, and try to cover many unique possibilities (at least 10). The top 10 diagnoses are:

**Example output.**
1. Sézary syndrome
2. Adult T-cell leukemia/lymphoma
3. Mycosis fungoides
4. Atopic dermatitis
5. Psoriasis
6. Drug reaction with eosinophilia and systemic symptoms (DRESS)
7. Graft-versus-host disease (GVHD)
8. Cutaneous T-cell lymphoma, not otherwise specified
9. Hypereosinophilic syndrome
10. Systemic lupus erythematosus (SLE)

### The AMIE user interface

The interface associated with AMIE, depicted in Extended Data Fig. 2, enables users to interact with the underlying model via text-based chat in the context of a given case description. In our study, the interface was pre-populated with a text-only representation of the HPI for a given case. Clinicians were asked to initiate the interaction by querying AMIE using a suggested prompt. Following this initial prompt and AMIE's response, clinicians were free to query the model using any additional follow-up questions, though clinicians were cautioned to avoid asking questions about information that had not already been presented in the case. A pilot study indicated that without such a warning, clinicians may ask questions about specific lab values or imaging leading to confabulations.

For a given question, the interface generated the response by querying AMIE using the following prompt template:

Read the case below and answer the question provided after the case.

Format your response in markdown syntax to create paragraphs and bullet points. Use '<br><br>' to start a new paragraph. Each paragraph should be 100 words or less. Use bullet points to list multiple options. Use '<br>*' to start a new bullet point. Emphasize important phrases like headlines. Use '**' right before and right after a phrase to emphasize it. There must be NO space in between '**' and the phrase you try to emphasize.

Case:[Case Text]
Question (suggested initial question is 'What are the top 10 most likely diagnoses and why (be precise)?'): [Question]
Answer:

## Experimental design

In order to comparatively evaluate AMIE's ability to generate a DDx alone and aid clinicians with their DDx generation we designed a two-stage reader study illustrated in Extended Data Fig. 3. Our study was designed to evaluate the assistive effect of AMIE for generalist clinicians (not specialists) who only have access to the case presentation and not the full case information (which would include the expert commentary on the DDx). The first stage of the study had a counterbalanced design with two conditions. Clinicians generated DDx lists first without assistance and then a second time with assistance, where the type of assistance varied by condition.

### Stage 1: Clinicians generate DDx with and without assistance.
Twenty U.S. board-certified internal medicine physicians (median years of experience: 9, mean: 11.5, s.d.: 7.24, range: 3–32) viewed the redacted case report, with access to the case presentation and associated figures and tables. They did this task in one of two conditions, based on random assignment.

Condition I: Search. The clinicians were first instructed to provide a list of up to ten diagnoses, with a minimum of three, based solely on review of the case presentation without using any reference materials (for example, books) or tools (for example, internet search). Following this, the clinicians were instructed to use internet search or other resources as desired (but not given access to AMIE) and asked to re-perform their DDx.

Condition II: AMIE. As with condition I, the clinicians were first instructed to provide a list of up to ten diagnoses, with a minimum of three, based solely on review of the case presentation without using any reference materials (for example, books) or tools (for example, internet search). Following this the clinicians were given access to AMIE and asked to re-perform their DDx. In addition to AMIE, clinicians could choose to use internet search or other resources if they wished.

For the assignment process, we formed ten pairs of two clinicians each, grouping clinicians with similar years of post-residency experience together. The set of all cases was then randomly split into ten partitions, and each clinician pair was assigned to one of the ten case partitions. Within each partition, each case was completed once in condition I by one of the two clinicians, and once in condition II by the other clinician. For each case, the assignment of which clinician among the pair was exposed to which of the two experimental conditions was randomized. Pairing clinicians with similar post-residency experience to complete the same case served to reduce variability between the two distinct experimental conditions.

### Stage 2. Specialists with full case information extract gold DDx and evaluate Stage 1 DDx
Nineteen U.S. board-certified specialist clinicians (median years of experience: 14, mean: 13.7, s.d.: 7.82, range: 4–38) were recruited from internal medicine ($n = 10$), neurology ($n = 3$), paediatrics ($n = 2$), psychiatry ($n = 1$), dermatology ($n = 1$), obstetrics ($n = 1$), and emergency medicine ($n = 1$). Their mean years of experience was 13.7 (s.d.: 7.82, range: 4–38). These specialists were aligned with the specialty of the respective CPC case, viewed the full case report and were asked to list at least five and up to ten differential diagnoses. Following this, they were asked to evaluate the five DDx lists generated in stage 1, including two DDx lists from condition 1 (DDx without assistance and DDx with Search assistance), two DDx lists from condition 2 (DDx without assistance and DDx with AMIE assistance) and the standalone AMIE DDx list. One specialist reviewed each case.

The specialists answered the following questions to evaluate the DDx lists:

The quality score developed by Bond et al.[15] and used by Kanjee et al.[1] is a differential score based on an ordinal five-point scale: 'How close did the differential diagnoses (DDx) come to including the final diagnosis?' The options were: 5, DDx includes the correct diagnosis; 4, DDx contains something that is very close, but not an exact match to the correct diagnosis; 3, DDx contains something that is closely related and might have been helpful in determining the correct diagnosis; 2, DDx contains something that is related, but unlikely to be helpful in determining the correct diagnosis; and 1, nothing in the DDx is related to the correct diagnosis.

An appropriateness score: 'How appropriate was each of the differential diagnosis lists from the different medical experts compared the differential list that you just produced?' The options to respond were on a Likert scale of 5 (very appropriate) to 1 (very inappropriate).

A comprehensiveness score: 'Using your differential diagnosis list as a benchmark/gold standard, how comprehensive are the differential lists from each of the experts?' The options to respond were: 4, the DDx contains all candidates that are reasonable; 3, the DDx contains most of the candidates but some are missing; 2, the DDx contains some of the candidates but a number are missing;' and 1, the DDx has major candidates missing.

Finally, specialists were asked to specify in which position of the DDx list the correct diagnosis was matched, in case it was included in the DDx at all.

**Clinician incentives.** Clinicians were recruited and remunerated by vendor companies at market rates based on speciality, without specific incentives such as diagnostic accuracy or other factors.

**Automated evaluation.** In addition to comparing against ground truth diagnosis and expert evaluation from clinicians, we also created an automated evaluation of the performance of the five DDxs using a language model-based metric. Such automated metrics are useful as human evaluation is time and cost-prohibitive for many experiments. We first extracted the (up to ten) individual diagnoses listed in each DDx. We leveraged minor text-processing steps via regular expressions to separate the outputs by newlines and strip any numbering before the diagnoses. Then we asked a medically fine-tuned language model, Med-PaLM 2[10], whether or not each of these diagnoses was the same as the ground truth diagnosis using the following prompt:

Is our predicted diagnosis correct (y/n)? Predicted diagnosis: [diagnosis], True diagnosis: [label]
Answer [y/n].

A diagnosis was marked as correct if the language model output 'y'.

We computed Cohen's kappa as a measure of agreement between human raters and automated evaluation with respect to the binary decision of whether a given diagnosis—that is, an individual item from a proposed DDx list—matched the correct final diagnosis. Cohen's kappa for this matching task was 0.631, indicating 'substantial agreement' between human raters and our automated evaluation method, per Landis & Koch[37].

## Qualitative interviews

Following the study we performed a semi-structured 30-min interviews with 5 of the generalist clinicians who participated in stage 1. Semi-structured interviews explored the following questions:
(1) How did you find the task of generating a DDx from the case report text?
(2) Think about how you used Internet search or other resources. How were these tools helpful or unhelpful?
(3) Think about how you used the AMIE. How was it helpful or unhelpful?
(4) Were there cases where you trusted the output of the search queries? Tell us more about the experience if so, such as types of cases, types of search results.
(5) Were there cases where you trusted the output of the LLM queries? Tell us more about the experience if so, such as types of cases, types of search results.
(6) Think about the reasoning provided by the LLM's interface? Where were they helpful? Where were they unhelpful?

(7) What follow-up questions did you find most helpful to ask the LLM?
(8) How much time does it take to get used to the LLM? How was it intuitive? How was it unintuitive?

We conducted a thematic analysis of notes from interviews taken by researchers during the interviews, employing an inductive approach to identify patterns (themes) within the data. Initial codes were generated through a line-by-line review of the notes, with attention paid to both semantic content and latent meaning. Codes were then grouped based on conceptual similarity, and refined iteratively. To enhance the trustworthiness of the analysis, peer debriefing was conducted within the team of researchers. Through discussion and consensus, the final themes were agreed upon.

## Reporting summary

Further information on research design is available in the Nature Portfolio Reporting Summary linked to this article.

## Data availability

The case reports used in this study are published and were licensed from the *New England Journal of Medicine*. We are not able to re-distribute the copyrighted material, but the case texts can be obtained from the journal.

## Code availability

AMIE is an LLM-based research AI system for diagnostic dialogue. We are not making the model code and weights open source owing to the safety implications of unmonitored use of such a system in medical settings. In the interest of responsible innovation, we will be working with research partners, regulators and providers to validate and explore safe onward uses of AMIE. For reproducibility, we have documented technical deep learning methods while keeping the paper accessible to a clinical and general scientific audience. Our work builds on PaLM 2, for which technical details have been described extensively in the technical report[7].

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

**Acknowledgements** This project was an extensive collaboration between many teams at Google Research and Google DeepMind. We thank A. Jain, R. Sayres, S. Lachgar, L. Winer, M. Shiels, B. Hatfield, S. W. Man, P. Singh, A. Um'rani, B. Green and P. Mansfield for their valuable insights and feedback during our research; A. Iurchenko for driving the design of the interactive user interface; M. Howell, M. Morris, C. Grade, K. DeSalvo, Z. Ghahramani, J. Manyika and J. Dean for their support during the course of this project; and the Massachusetts Medical Society group for the support and partnership.

**Author contributions** D.M., M.S., T.T., A. Palepu, A. Pathak, J.S., A.K. and V.N. contributed to the conception and design of the work. D.M., M.S., T.T., A. Palepu, A.W., Y.S., K.K., J.S., A.K. and V.N. contributed to the data acquisition and curation. T.T., A. Palepu, D.M., M.S. and K.S. contributed to the technical implementation. L.H., Y.C., Y.L., S.S.M., S. Prakash and A. Pathak provided technical and infrastructure guidance. J.S and A.K. provided clinical inputs to the study. D.M., M.S., T.T., A. Palepu, A.W., J. Garrison, K.S., Y.S., S.A., L.H., Y.C., Y.L., S.S.M., S. Prakash, A. Pathak, C.S., S. Patel, D.R.W., E.D., J. Gottweis, J.B., K.C., G.S.C., Y.M., J.S., A.K. and V.N. contributed to the drafting and revising of the manuscript.

**Competing interests** This study was funded by Alphabet Inc. and/or a subsidiary thereof ('Alphabet'). All authors are employees of Alphabet and may own stock as part of the standard compensation.

**Additional information**
**Correspondence and requests for materials** should be addressed to Daniel McDuff, Mike Schaekermann, Jake Sunshine, Alan Karthikesalingam or Vivek Natarajan.

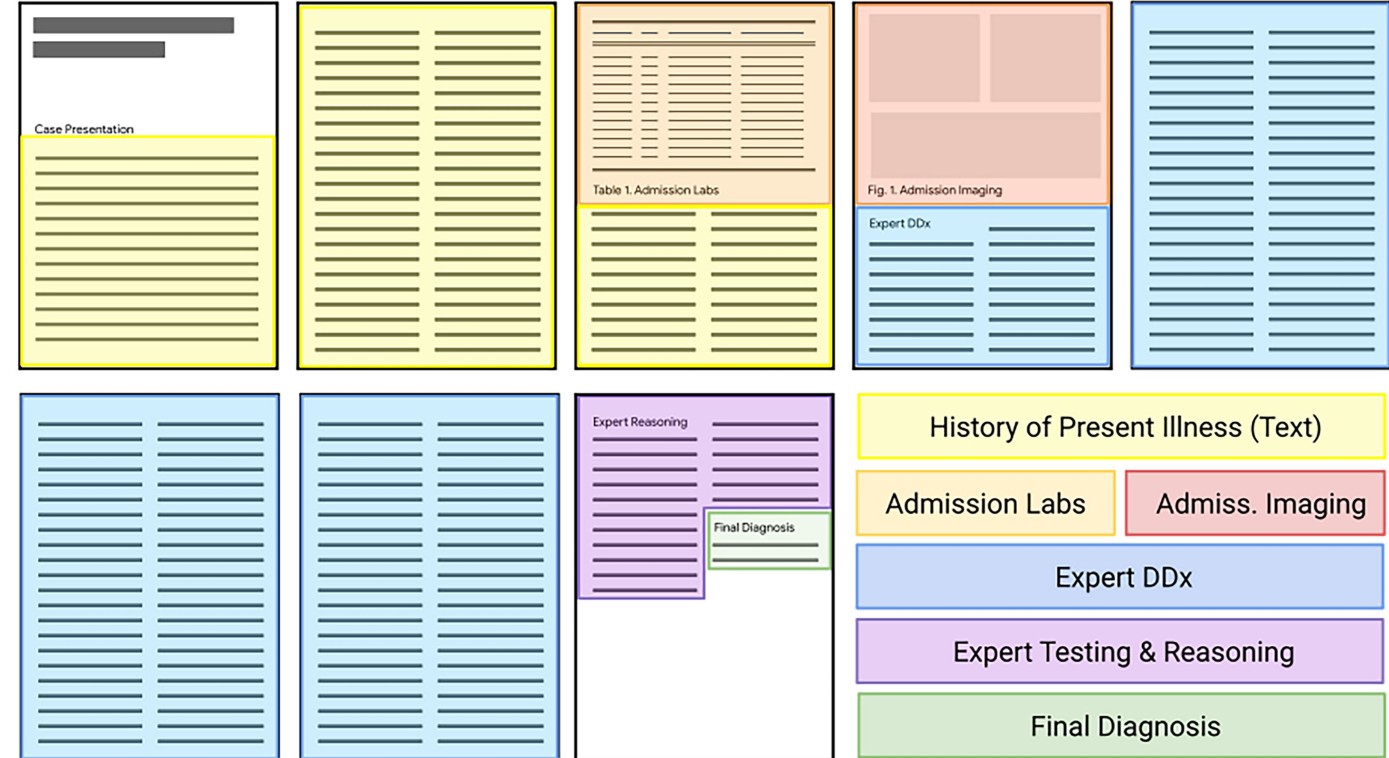

**Extended Data Fig. 1 | NEJM Clinicopathological Conference Case Reports.** History of Present Illness, Admission Labs and Admission Imaging sections were included in the redacted version presented to generalist clinicians for producing a DDx. The LLM had access to only the History of Present Illness. Specialist clinicians evaluating the quality of the DDx had access to the full (unredacted) case report including the expert differential discussion.

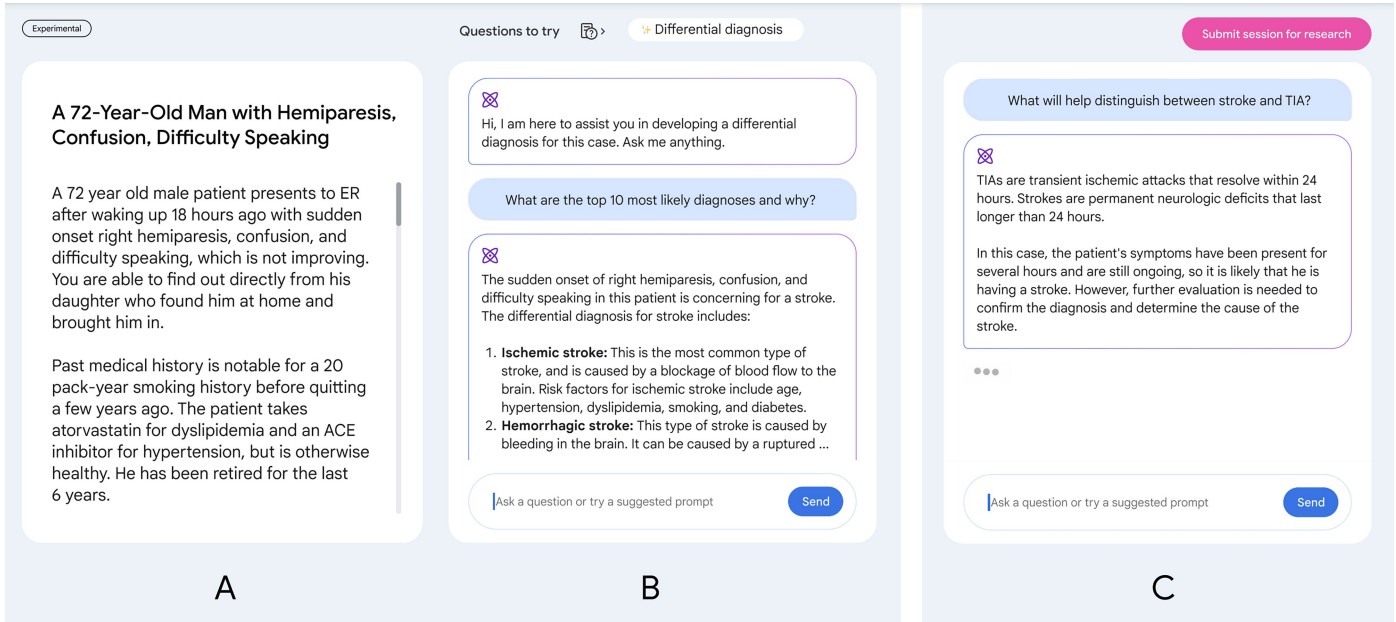

**Extended Data Fig. 2 | The AMIE User Interface.** The history of the present illness (text only) was pre-populated in the user interface (A) with an initial suggested prompt to query the LLM (B). Following this prompt and response, the user was free to enter any additional follow-up questions (C). The case shown in this figure is a mock case selected for illustrative purposes only.

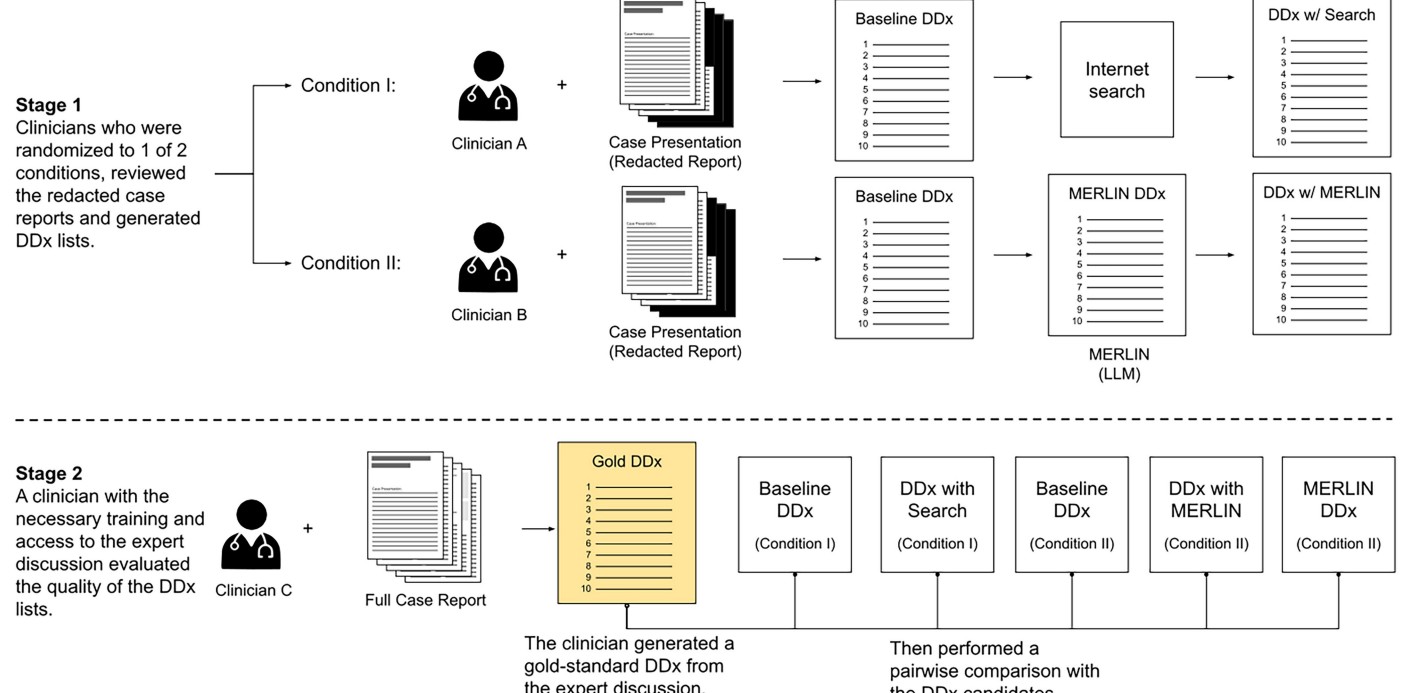

**Extended Data Fig. 3 | Experimental Design.** To evaluate the LLM's ability to generate DDx lists and aid clinicians with their DDx generation, we designed a two-stage reader study. First, clinicians with access only to the case presentation completed DDx lists without using any assistive tools. Second, the clinicians completed DDx lists with access *either* to Search engines and other resources (Condition I), or to LLM in addition to these tools (Condition II). Randomization was employed such that every case was reviewed by two different clinicians, one with LLM assistance and one without. In Condition II the clinician was given a suggested initial prompt to use in the LLM interface and was then free to try any other questions. These DDx lists were then evaluated by a specialist who had access to the full case and expert commentary on the differential diagnosis, but who was blinded to whether and what assistive tool was used.

# Reporting Summary

## Statistics

For all statistical analyses, confirm that the following items are present in the figure legend, table legend, main text, or Methods section.

| n/a | Confirmed | |
|---|---|---|
| ☐ | ☒ | The exact sample size (*n*) for each experimental group/condition, given as a discrete number and unit of measurement |
| ☐ | ☒ | A statement on whether measurements were taken from distinct samples or whether the same sample was measured repeatedly |
| ☐ | ☒ | The statistical test(s) used AND whether they are one- or two-sided *Only common tests should be described solely by name; describe more complex techniques in the Methods section.* |
| ☐ | ☒ | A description of all covariates tested |
| ☐ | ☒ | A description of any assumptions or corrections, such as tests of normality and adjustment for multiple comparisons |
| ☐ | ☒ | A full description of the statistical parameters including central tendency (e.g. means) or other basic estimates (e.g. regression coefficient) AND variation (e.g. standard deviation) or associated estimates of uncertainty (e.g. confidence intervals) |
| ☐ | ☒ | For null hypothesis testing, the test statistic (e.g. *F*, *t*, *r*) with confidence intervals, effect sizes, degrees of freedom and *P* value noted *Give P values as exact values whenever suitable.* |
| ☒ | ☐ | For Bayesian analysis, information on the choice of priors and Markov chain Monte Carlo settings |
| ☒ | ☐ | For hierarchical and complex designs, identification of the appropriate level for tests and full reporting of outcomes |
| ☐ | ☒ | Estimates of effect sizes (e.g. Cohen's *d*, Pearson's *r*), indicating how they were calculated |

*Our web collection on statistics for biologists contains articles on many of the points above.*

## Software and code

Policy information about availability of computer code

| Data collection | The algorithms and scripts were implemented using Python. |
|---|---|
| Data analysis | The data analyses scripts were implemented in Python.  We will not be able to open source the LLMs used in this study. |

For manuscripts utilizing custom algorithms or software that are central to the research but not yet described in published literature, software must be made available to editors and reviewers. We strongly encourage code deposition in a community repository (e.g. GitHub). See the Nature Portfolio guidelines for submitting code & software for further information.

## Data

Policy information about availability of data

All manuscripts must include a data availability statement. This statement should provide the following information, where applicable:
- Accession codes, unique identifiers, or web links for publicly available datasets
- A description of any restrictions on data availability
- For clinical datasets or third party data, please ensure that the statement adheres to our policy

We have provided the set of case IDs and the diagnoses generated by the models and clinicians as supplemental material with our submission.  We have also provided instructions about how to access the model end-point for testing.

# Human research participants

Policy information about studies involving human research participants and Sex and Gender in Research.

| | |
|---|---|
| Reporting on sex and gender | n/a |
| Population characteristics | Nothing to add. |
| Recruitment | n/a |
| Ethics oversight | n/a |

Note that full information on the approval of the study protocol must also be provided in the manuscript.

# Field-specific reporting

Please select the one below that is the best fit for your research. If you are not sure, read the appropriate sections before making your selection.

☒ Life sciences  ☐ Behavioural & social sciences  ☐ Ecological, evolutionary & environmental sciences

For a reference copy of the document with all sections, see nature.com/documents/nr-reporting-summary-flat.pdf

# Life sciences study design

All studies must disclose on these points even when the disclosure is negative.

| | |
|---|---|
| Sample size | The study included 302 published case reports from the New England Journal of Medicine. |
| Data exclusions | All valid case reports with differential diagnoses were used. |
| Replication | The evaluations were performed by clinical specialists. |
| Randomization | The study arms (AMIE and Search) were randomized. The cases were also randomized amongst clinicians but within clinical specialties. |
| Blinding | The clinicians were not told which study arm they were exposed to in each case or which study condition they were evaluating responses from. |

# Reporting for specific materials, systems and methods

We require information from authors about some types of materials, experimental systems and methods used in many studies. Here, indicate whether each material, system or method listed is relevant to your study. If you are not sure if a list item applies to your research, read the appropriate section before selecting a response.

### Materials & experimental systems

| n/a | Involved in the study |
|---|---|
| ☐ ☐ | Antibodies |
| ☐ ☐ | Eukaryotic cell lines |
| ☐ ☐ | Palaeontology and archaeology |
| ☐ ☐ | Animals and other organisms |
| ☐ ☐ | Clinical data |
| ☐ ☐ | Dual use research of concern |

### Methods

| n/a | Involved in the study |
|---|---|
| ☐ ☐ | ChIP-seq |
| ☐ ☐ | Flow cytometry |
| ☐ ☐ | MRI-based neuroimaging |

## Antibodies

| | |
|---|---|
| Antibodies used | *Describe all antibodies used in the study; as applicable, provide supplier name, catalog number, clone name, and lot number.* |
| Validation | *Describe the validation of each primary antibody for the species and application, noting any validation statements on the manufacturer's website, relevant citations, antibody profiles in online databases, or data provided in the manuscript.* |

# Eukaryotic cell lines

Policy information about cell lines and Sex and Gender in Research

| | |
|---|---|
| Cell line source(s) | *State the source of each cell line used and the sex of all primary cell lines and cells derived from human participants or vertebrate models.* |
| Authentication | *Describe the authentication procedures for each cell line used OR declare that none of the cell lines used were authenticated.* |
| Mycoplasma contamination | *Confirm that all cell lines tested negative for mycoplasma contamination OR describe the results of the testing for mycoplasma contamination OR declare that the cell lines were not tested for mycoplasma contamination.* |
| Commonly misidentified lines (See ICLAC register) | *Name any commonly misidentified cell lines used in the study and provide a rationale for their use.* |

# Palaeontology and Archaeology

| | |
|---|---|
| Specimen provenance | *Provide provenance information for specimens and describe permits that were obtained for the work (including the name of the issuing authority, the date of issue, and any identifying information). Permits should encompass collection and, where applicable, export.* |
| Specimen deposition | *Indicate where the specimens have been deposited to permit free access by other researchers.* |
| Dating methods | *If new dates are provided, describe how they were obtained (e.g. collection, storage, sample pretreatment and measurement), where they were obtained (i.e. lab name), the calibration program and the protocol for quality assurance OR state that no new dates are provided.* |

☐ Tick this box to confirm that the raw and calibrated dates are available in the paper or in Supplementary Information.

| | |
|---|---|
| Ethics oversight | *Identify the organization(s) that approved or provided guidance on the study protocol, OR state that no ethical approval or guidance was required and explain why not.* |

Note that full information on the approval of the study protocol must also be provided in the manuscript.

# Animals and other research organisms

Policy information about studies involving animals; ARRIVE guidelines recommended for reporting animal research, and Sex and Gender in Research

| | |
|---|---|
| Laboratory animals | *For laboratory animals, report species, strain and age OR state that the study did not involve laboratory animals.* |
| Wild animals | *Provide details on animals observed in or captured in the field; report species and age where possible. Describe how animals were caught and transported and what happened to captive animals after the study (if killed, explain why and describe method; if released, say where and when) OR state that the study did not involve wild animals.* |
| Reporting on sex | *Indicate if findings apply to only one sex; describe whether sex was considered in study design, methods used for assigning sex. Provide data disaggregated for sex where this information has been collected in the source data as appropriate; provide overall numbers in this Reporting Summary. Please state if this information has not been collected. Report sex-based analyses where performed, justify reasons for lack of sex-based analysis.* |
| Field-collected samples | *For laboratory work with field-collected samples, describe all relevant parameters such as housing, maintenance, temperature, photoperiod and end-of-experiment protocol OR state that the study did not involve samples collected from the field.* |
| Ethics oversight | *Identify the organization(s) that approved or provided guidance on the study protocol, OR state that no ethical approval or guidance was required and explain why not.* |

Note that full information on the approval of the study protocol must also be provided in the manuscript.

# Clinical data

Policy information about clinical studies

All manuscripts should comply with the ICMJE guidelines for publication of clinical research and a completed CONSORT checklist must be included with all submissions.

| | |
|---|---|
| Clinical trial registration | *Provide the trial registration number from ClinicalTrials.gov or an equivalent agency.* |
| Study protocol | *Note where the full trial protocol can be accessed OR if not available, explain why.* |
| Data collection | *Describe the settings and locales of data collection, noting the time periods of recruitment and data collection.* |

| Outcomes | *Describe how you pre-defined primary and secondary outcome measures and how you assessed these measures.* |
|---|---|

# Dual use research of concern

Policy information about dual use research of concern

## Hazards

Could the accidental, deliberate or reckless misuse of agents or technologies generated in the work, or the application of information presented in the manuscript, pose a threat to:

| No | Yes | |
|---|---|---|
| ☐ | ☐ | Public health |
| ☐ | ☐ | National security |
| ☐ | ☐ | Crops and/or livestock |
| ☐ | ☐ | Ecosystems |
| ☐ | ☐ | Any other significant area |

## Experiments of concern

Does the work involve any of these experiments of concern:

| No | Yes | |
|---|---|---|
| ☐ | ☐ | Demonstrate how to render a vaccine ineffective |
| ☐ | ☐ | Confer resistance to therapeutically useful antibiotics or antiviral agents |
| ☐ | ☐ | Enhance the virulence of a pathogen or render a nonpathogen virulent |
| ☐ | ☐ | Increase transmissibility of a pathogen |
| ☐ | ☐ | Alter the host range of a pathogen |
| ☐ | ☐ | Enable evasion of diagnostic/detection modalities |
| ☐ | ☐ | Enable the weaponization of a biological agent or toxin |
| ☐ | ☐ | Any other potentially harmful combination of experiments and agents |

# ChIP-seq

## Data deposition

☐ Confirm that both raw and final processed data have been deposited in a public database such as GEO.

☐ Confirm that you have deposited or provided access to graph files (e.g. BED files) for the called peaks.

| Data access links<br>*May remain private before publication.* | *For "Initial submission" or "Revised version" documents, provide reviewer access links. For your "Final submission" document, provide a link to the deposited data.* |
|---|---|
| Files in database submission | *Provide a list of all files available in the database submission.* |
| Genome browser session<br>(e.g. UCSC) | *Provide a link to an anonymized genome browser session for "Initial submission" and "Revised version" documents only, to enable peer review. Write "no longer applicable" for "Final submission" documents.* |

## Methodology

| Replicates | *Describe the experimental replicates, specifying number, type and replicate agreement.* |
|---|---|
| Sequencing depth | *Describe the sequencing depth for each experiment, providing the total number of reads, uniquely mapped reads, length of reads and whether they were paired- or single-end.* |
| Antibodies | *Describe the antibodies used for the ChIP-seq experiments; as applicable, provide supplier name, catalog number, clone name, and lot number.* |
| Peak calling parameters | *Specify the command line program and parameters used for read mapping and peak calling, including the ChIP, control and index files used.* |
| Data quality | *Describe the methods used to ensure data quality in full detail, including how many peaks are at FDR 5% and above 5-fold enrichment.* |
| Software | *Describe the software used to collect and analyze the ChIP-seq data. For custom code that has been deposited into a community repository, provide accession details.* |

# Flow Cytometry

## Plots

Confirm that:

☐ The axis labels state the marker and fluorochrome used (e.g. CD4-FITC).

☐ The axis scales are clearly visible. Include numbers along axes only for bottom left plot of group (a 'group' is an analysis of identical markers).

☐ All plots are contour plots with outliers or pseudocolor plots.

☐ A numerical value for number of cells or percentage (with statistics) is provided.

## Methodology

| | |
|---|---|
| Sample preparation | *Describe the sample preparation, detailing the biological source of the cells and any tissue processing steps used.* |
| Instrument | *Identify the instrument used for data collection, specifying make and model number.* |
| Software | *Describe the software used to collect and analyze the flow cytometry data. For custom code that has been deposited into a community repository, provide accession details.* |
| Cell population abundance | *Describe the abundance of the relevant cell populations within post-sort fractions, providing details on the purity of the samples and how it was determined.* |
| Gating strategy | *Describe the gating strategy used for all relevant experiments, specifying the preliminary FSC/SSC gates of the starting cell population, indicating where boundaries between "positive" and "negative" staining cell populations are defined.* |

☐ Tick this box to confirm that a figure exemplifying the gating strategy is provided in the Supplementary Information.

# Magnetic resonance imaging

## Experimental design

| | |
|---|---|
| Design type | *Indicate task or resting state; event-related or block design.* |
| Design specifications | *Specify the number of blocks, trials or experimental units per session and/or subject, and specify the length of each trial or block (if trials are blocked) and interval between trials.* |
| Behavioral performance measures | *State number and/or type of variables recorded (e.g. correct button press, response time) and what statistics were used to establish that the subjects were performing the task as expected (e.g. mean, range, and/or standard deviation across subjects).* |

## Acquisition

| | |
|---|---|
| Imaging type(s) | *Specify: functional, structural, diffusion, perfusion.* |
| Field strength | *Specify in Tesla* |
| Sequence & imaging parameters | *Specify the pulse sequence type (gradient echo, spin echo, etc.), imaging type (EPI, spiral, etc.), field of view, matrix size, slice thickness, orientation and TE/TR/flip angle.* |
| Area of acquisition | *State whether a whole brain scan was used OR define the area of acquisition, describing how the region was determined.* |

Diffusion MRI        ☐ Used          ☐ Not used

## Preprocessing

| | |
|---|---|
| Preprocessing software | *Provide detail on software version and revision number and on specific parameters (model/functions, brain extraction, segmentation, smoothing kernel size, etc.).* |
| Normalization | *If data were normalized/standardized, describe the approach(es): specify linear or non-linear and define image types used for transformation OR indicate that data were not normalized and explain rationale for lack of normalization.* |
| Normalization template | *Describe the template used for normalization/transformation, specifying subject space or group standardized space (e.g. original Talairach, MNI305, ICBM152) OR indicate that the data were not normalized.* |
| Noise and artifact removal | *Describe your procedure(s) for artifact and structured noise removal, specifying motion parameters, tissue signals and physiological signals (heart rate, respiration).* |

| Volume censoring | *Define your software and/or method and criteria for volume censoring, and state the extent of such censoring.* |

## Statistical modeling & inference

| Model type and settings | *Specify type (mass univariate, multivariate, RSA, predictive, etc.) and describe essential details of the model at the first and second levels (e.g. fixed, random or mixed effects; drift or auto-correlation).* |
| Effect(s) tested | *Define precise effect in terms of the task or stimulus conditions instead of psychological concepts and indicate whether ANOVA or factorial designs were used.* |

Specify type of analysis: ☐ Whole brain ☐ ROI-based ☐ Both

| Statistic type for inference (See Eklund et al. 2016) | *Specify voxel-wise or cluster-wise and report all relevant parameters for cluster-wise methods.* |
| Correction | *Describe the type of correction and how it is obtained for multiple comparisons (e.g. FWE, FDR, permutation or Monte Carlo).* |

## Models & analysis

| n/a | Involved in the study |
| --- | --- |
| ☐ | ☐ Functional and/or effective connectivity |
| ☐ | ☐ Graph analysis |
| ☐ | ☐ Multivariate modeling or predictive analysis |

| Functional and/or effective connectivity | *Report the measures of dependence used and the model details (e.g. Pearson correlation, partial correlation, mutual information).* |
| Graph analysis | *Report the dependent variable and connectivity measure, specifying weighted graph or binarized graph, subject- or group-level, and the global and/or node summaries used (e.g. clustering coefficient, efficiency, etc.).* |
| Multivariate modeling and predictive analysis | *Specify independent variables, features extraction and dimension reduction, model, training and evaluation metrics.* |

