## [Peer Review File · Nature]

Towards Accurate Differential Diagnosis with Large Language Models

Corresponding Author: Dr Daniel McDuff

Parts of this Peer Review File have been redacted as indicated to maintain confidentiality.

Version 1:

Reviewer comments:

Referee #1

(Remarks to the Author)

My summary of the key results:

The article explores the performance of AMIE in differential diagnosis of 302 NEJM Clinicopathological conferences with comparison to 20 board-certified internists. The internists all perform the differentials unaided and are afterwards allowed aid by either search only or AMIE (and search). Specialists in the appropriate fields review the results based on Top-n-accuracy, comprehensiveness of DDX lists, quality of the lists and appropriateness. The article finds that unaided, the internists are outperformed on every metric. Clinicians with assistance outperform unaided clinicians, with clinicians aided by AMIE performing the best. AMIE alone outperforms all clinician groups.

I find the article to be original and of great significance. Whilst as the article states itself, LLMs have already been shown to perform well on NEJM cases, the performance of LLMs has yet to be compared to verified doctors. This article presents compelling evidence that LLMs, at least AMIE, can indeed outperform board-certified internists.

Some points and suggested improvements:

Regarding methods:

Overall, I appreciate the methods of the article. From a technical standpoint they seem valid overall, and the authors has generated and presented a large dataset. I appreciate the contamination analysis as well, as I imagine this would otherwise be a point of contention.

1. I choose the word 'seem' as I do not have access to AMIE (nor even Med-PaLM2). Access requests have been futile. I cannot verify the data the article presents; however, I can say that with my experience with other LLM in a similar dataset with similar prompts, the data does not at all seem unrealistic, especially for an LLM tuned in medicine.
2. I would have liked to see some data on consistency – the article would benefit from analysis on repeated runs.
3. I wonder about the "top 10 diagnoses" approach. Considering 10 diagnoses would generally be a lot in real life and I have doubts about the clinical relevance of this. I don't know if there are any data on this question, but this very article finds that the median is 6 diagnoses for unassisted doctors. This warrants discussion in my opinion.
4. In relation to this, I am concerned about the blinding of the human raters. If the doctors output a mean 6 diagnoses (unaided) and AMIE always outputs 10 diagnoses, it seems there is a risk of being able to figure out which is which. The article makes no mention of how the doctors were instructed to write their diagnoses, so spelling mistakes, abbreviations and differences in connotation could exacerbate this issue. This seems to be a potential point of bias and any potential conflicts of interest. Elaboration would improve the article.
 - a. In relation to this, how were the participating doctors (diagnosers and evaluators) incentivized? I think the article should state this for transparency.
5. The article compares a group that has access to labs and one that does not. I am aware of model limitations, but doctors using the model in real life would probably be tempted to provide labs in the prompt, meaning it may not be a realistic comparison either. GPT-4 seems to perform adequately with labs. If possible, I believe it should be added for a more complete comparison, but this might not be feasible.

Regarding statistics, I am no expert on statistics. I did have a few concerns regarding statistics that I ran by our statistician. He offered the following advice:

6. Page 8: "The mean appropriateness score of the LLM (4.34)[...]" - The data most likely does not follow a normal distribution (as they can only be a few whole integers), so he would recommend a signrank test.

7. Page 9: "The mean appropriateness score after assistance [...]" – Same as above.

These are my own comments regarding statistics:

8. Did the contamination analysis change anything from significant to insignificant? It is vaguely described, only specifically mentioning significance for clinicians assisted by LLM vs without LLM.

9. I would have liked at least a bit of data on the qualitative analysis, I think this is a very interesting aspect, but perhaps it extends outside the scope of this article.

Regarding conclusions. The overall conclusion that AMIE outperforms unassisted doctors seems very well supported. The conclusion that aided doctors (both groups) outperforms unassisted doctors also seem well supported by article.

10. The article concludes that doctors using AMIE outperformed doctors using search on top-n-diagnoses. I would like to challenge the extent to which it is valid based on the clinical relevancy of this conclusion. The difference does seem arise for sure (only graph shown) until top 5 diagnoses by human evaluation or top 6 diagnoses by automated evaluation. The article reports that the unaided doctor puts a median of 6 diagnoses in their DDx in these case challenge scenarios.

11. Regarding human vs automatic evaluation, the article concludes that there is reasonable agreement between the two. While the two graphs do look somewhat similar, the data is insufficient to support the conclusion (no supplementary either).

a. I ran this bullet by our statistician as well, who also reckon the statement is too vague for the shown data.

12. This conclusion is later used to build the basis for concluding that AMIE outperforms GPT-4 (based on automatic evaluation). However, based on the aforementioned lack of data supporting this, I believe this conclusion is put into question. This conclusion is also put into question by the contrast to the results of GPT-4 in Kanjee et al. rated by humans.

13. The "time spent" analysis is in my opinion not rooted in reality and I question the validity, as it neglects the fact that the presented case was pre-populated, something a doctor using an LLM would have to write themselves from the EMRs.

Despite being a proponent of AI I use search a lot more than LLM in my clinical practice due to this time factor alone.

Minor points

- The article mentions supplementary data which is missing, as far as I can tell.

- I stumbled upon a few incomplete sentences missing words, which should not be present in a high-profile article such as this. Example page 8 line 5 of main text (following "The mean appropriateness [...]") and page 11 line 6 of main text (sentence: ". performs better"). A read-through is in order.

- The article does not mention specifically how many specialists evaluate each case. I assume 1.

- I believe figure 3 and the text does not match completely? AMIE DDx is put as "Condition II", but in condition II the article describes that the AMIE DDx is only a suggested first prompt, suggesting that the doctors in condition II could put something else (e.g. top 5 or something unrelated). The LLM was assessed on its own, not as part of condition II, correct?

- Might be a language barrier issue, but "The DDx contains all candidates that are reasonable" is confusing to me, I hope it can be clarified a bit. Based on the category 2 and 3 title it seems that it would mean that all the reasonable diagnoses from the Gold DDx list is included? But why would the Gold DDx list include unreasonable diagnoses?

I found the references appropriate, but "Towards Conversational Diagnostic AI" by Tao Tu et al. was not mentioned despite exploring the same model and by some of the same authors. (Was released on Arxiv after this one)

Regarding clarity and context, I found the entire article, including abstract, to be clear, concise and appropriate. I particularly enjoyed the discussion.

I want to extend my congratulations to the authors for a tremendous work, I was happy to be allowed to review it, and I want to assure the authors that I believe this a very high-quality article with major impact, despite my comments above. I want to apologize for the length, but it feels like an important article and an important journal. I want to thank Nature for the chance to review this article, it has been an honor.

(Remarks on code availability)

Somewhat;

As mentioned in reviewer comments, AMIE and Med-PaLM2 are not accessible to me.

Thus, I cannot test the results of the prompts in the actual API.

However, I've done very extensive testing of similar prompts and also tested their prompt in GPT-4. Whilst GPT-4 is not the same and LLMs are not interchangeable (even different versions of the same LLM might not give the same response), the prompts SHOULD work in an advanced LLM trained in medicine. I've used the publicly available version of Gemini which functions at a high level, so I have no reason to believe Google would not be able to tune an LLM to medicine.

In so many words, based on my experience with LLMs this would work, and I believe it would work with other advanced LLMs as well.

HOWEVER, if you want a guarantee that AMIE can actually answer what they say it can (e.g. test the prompts in the actual program), you will have to pressure Google into providing access. I think your name carries a lot more sway than mine in this regard :-)

I've previously requested access from Google to their LLMs without luck and I have done so again due to this review, although I've not told them I wanted it for this review specifically as I did not want to break any confidentiality rules, but maybe that would help.

Referee #2

(Remarks to the Author)

The paper examines the performance of “AMIE, an LLM model optimized for differential diagnosis, alongside a user interface allowing clinicians to interact with the model for improving clinical diagnostic reasoning” on the task of creating differential diagnosis lists from 302 NEJM CPC cases published in the past decade. AMIE is based on the Large PaLM 2 model, fine-tuned with a variety of available medical QA datasets, which were also used in this group’s 2023 paper on medical question answering, plus apparently some others not described (“The datasets used included ...”).

The headline result is that after a panel of domain experts determined the gold standard diagnoses for the CPC cases, AMIE on its own included the correct diagnosis among its top 10 generated ones in 59.1% of the cases and nailed the top diagnosis in 29%. Human experiments with 20 board-certified physicians with a median of 9 years’ experience showed that AMIE’s performance on the top-10 task was significantly better than the doctors’, whose top-10 accuracy was only 36.1%, or 44.4% when encouraged to use Web search. Oddly, their performance when using AMIE, whose goal is to improve their abilities, was 51.7%, considerably lower than what AMIE achieved by itself, though much better than their unassisted performance. One of my concerns with this paper is that these results suggest a take-away lesson that we should supplant physicians by AMIE-like systems for clinical diagnosis: “the proposed LLM outperformed an unassisted board-certified physician in both top-1 and top-n performance”. To be fair, the authors do not suggest this, but I suspect that many readers will settle on that conclusion, driven by the reported results. I wish the paper included a serious attempt to understand why there is this gap. Previous claims for earlier physician assistance systems have often argued that the combination of a physician and a computational model do outperform either one by itself, yet this paper’s results do not conform to that expectation. The diagram in Figure 6 summarizes the aggregate effect of adding search or LLM, but it would be extremely helpful to investigate what actually changes in the doctors’ minds as they incorporate this additional information. What sorts of cases are helped by these decision support tools and what are harmed? How does the difficulty of the case (however that is judged) affect this? I’d be surprised if it were just random.

One speculation I can offer for this surprising result is that the experiments they ran included two conditions for generating the DDx list, the first using no reference tools, followed by encouragement to use internet search or “other resources” (?), followed by use of the LLM, search, etc. From what we know about “anchoring” phenomena in human cognition, perhaps this ordering allows the doctors to fixate on their initial opinion and to show reluctance to accept apparently better advice fully. Unfortunately, it would probably take a much large sample of subjects to show significant results among the conditions if the experiments exposed each subject to just one actual condition (review of HPI, review with search, review with LLM).

The superior performance of the LLM is especially surprising since AMIE was given only the HPI of the CPC cases to work from, whereas the physicians also received admission labs and imaging studies. Because these are often sources of decisive information, I would have thought the physician performance might be better, given their access to more data. The paper says that such data were “sometimes included in the case description”—a rather vague explanation. Without some further insight into this question, I am left with the suspicion that the authors (and I) are missing something important.

Performance evaluation in the paper is based not only on top-n accuracy but also by asking experts to provide Likert scale-like evaluations of quality, comprehensiveness, and appropriateness. Although such evaluations can tease out more subtle criteria, such as the clinical importance of diagnoses that may have been missed or inappropriately included, my experience makes me suspicious of such “did you like it” methods. For example, Figure 4 shows that the “clinician unassisted” data in either the search or AMIE condition receive notably different distributions of judgments on each of the three scores, despite the fact that the first task in each condition is that unassisted judgment. I would expect similar results, at least if the subject populations were homogeneous. These results are, nevertheless, roughly in line with the top-n quantitative results.

I was confused by the description of their automated evaluation, which generally assigned slightly lower top-n scores to AMIE and slightly higher scores for each condition that includes the judgment of a clinician. From the brief description, it appears that Med-PaLM 2 was used to determine the correctness of each of the up to 10 diagnoses generated by AMIE or by the clinicians with various assistance tools. I cannot figure out what the relationship is between Med-PaLM 2 and the LLM included in AMIE. Are these actually the same? If not, what is the difference between them? If they are the same and the LLM has reasonable internal consistency, why would it judge its own top-1 performance to be only ~27% accurate (Fig 5 right)? I am generally concerned that we don’t understand why LLMs can do the kinds of reasoning they seem capable of, so insights would be valuable into this apparent disagreement between an LLM and itself (or perhaps a sibling).

I was happy to see a significant section on contamination analysis, which is an obvious concern in such studies. For example, the NY Times is apparently suing OpenAI for including copyrighted material in its training corpus, using experimental results that demonstrate that prompting GPT with an initial fragment of an article can pretty well reproduce the

entire article. Thus, it is natural to want to know the extent to which the DDx that is calculated by AMIE is simply being regurgitated from something its LLM has ingested previously. I am unsure whether the 512-character overlap analysis was done only on the medical QA data used to fine-tune PaLM 2 or if it applied to the vastly larger training data used in creating PaLM 2 as well. If only done on the fine tuning data, then that comparison is not very convincing.

The sentence's meaning "We performed an additional overlap analysis on articles less than, or equal to, 512-characters overlap (N = 249)" is unclear to me. Single-character overlap would presumably find overlap for all N = 302 cases, so what degree of overlap yielded 249? I could imagine other methods of searching for overlap that would not demand character-by-character equality. For example, would sentence or paragraph level computed embedding vectors from each source (or fine-tuning) corpus overlaid on similar vectors for cases yield insights into close paraphrases of the CPCs in other data?

Of minor note, there is a broken sentence in section 6.4. The brief comparison with GPT-4 shows better performance for top-2 or greater, but the GPT-4 results in fact appear (Fig 7) slightly better for top-1 when judged by either Med-PaLM 2 or GPT 4

The discussion helps to add nuance to the report and points out various limitations, including the unusual difficulty of NEJM CPC cases, the very limited investigation of HCI issues in this work, and the pragmatics of where and how such tools could be used to improve clinical practice.

I think one of the important products of the reported research is the determination of gold standard diagnosis judgments for the 302 NEJM cases they used. The Data Availability section reasonably argues that they are unable to disseminate the copyrighted text, but to make their study reproducible and to help advance the field, I would strongly recommend that they publish those judgments so everyone's research can benefit.

(Remarks on code availability)

Version 3:

Reviewer comments:

Referee #1

(Remarks to the Author)

I want thank the authors for a very comprehensive rebuttal of high quality. Going through every case again with labs (for example) will likely have taken quite a bit of work. Very well done.

My summary of the article remains the same, and as does the previous comments on the manuscript as a whole. It remains an important article and remains to my knowledge the first article that has provided compelling proof that LLMs indeed have the potential to out-perform physicians in clinical cases and enhance the performance of clinicians themselves.

In my opinion these findings show the future of our healthcare systems in the western world where we will have extreme demographic challenges. I think this article is an important piece of that.

I find that all my major concerns have been very well addressed in the rebuttal and in the renewed manuscript.

The approach is valid, the quality of the data is high and as high as can be expected - it requires quite a bit of resources to gather data like this from 20 doctors. The presentation of the data has improved quite a bit and I think the uncertainties regarding statistics that I presented have been addressed well.

I think the conclusions are well supported overall.

I only have two minor point that I think should be adressed prior to publication. This has occured ot me after having tested the model (AMIE) myself:

A.
In the article, it is mentioned a few times that this model outperforms GPT-4, also refered to as a state-of-the-art model. This may or may not be valid. It is based on the results by Kanjee et al, who used GPT-4 to solve cases. However, I would argue that as of November 2024, this conclusion that it outperforms "GPT-4" is no longer valid based alone on that it performed better than the results found by Kanjee. That is, it lacks support beyond just out-performing the results by Kanjee et al. The reason for this is that GPT-4 has been updated many times since Kanjee et al made their article. I have checked the article, and they unfortunately do not mention which model they used, however they do mention that the model has a learning cutoff of September 2021 which combined with the publication date, leads me to believe that it is GPT-4-0413 - Either way, it is not the current model and even for OpenAI internally it would not be considered state-of-the-art anymore.

I think the easiest way to alleviate this is simply to mention clarify this point in the article. Other options would be removing the comparison or actually testing it against the current model. I think this is an important point as OpenAI obviously represents Googles competition.

Very minor, but:

B. In the supplementary data, some cases do not have any data for the AMIE or do not contain 10 diagnoses even though that is the prompted answer. I assume this is a simple oversight. (i.e. the lines are just empty). Examples include Case 12-2023 and 29-2015

Once these points are addressed, I believe the article warrants publication.

I want to congratulate the authors on a tremendous effort. Great work, the manuscript was a pleasure to review.

(Remarks on code availability)

The attached supplementary data extract is consistent with what I would expect.

I have tested the model. The results are reproducible to the extent that is reasonable. The model seems to have a non-zero temperature, which means that it will not always output the exact same output, but in general it seems consistent with what has been shown in the supplementary.

Without testing extensively I cannot know if it would outperform GPT-4 as claimed due to the reasons mentioned above.

Testing that seems outside the scope of a reviewer.

As a remark for Nature, I would say that this has been a perfectly adequate way of handling this sort of testing (Where there is confidentiality issues but still a need for a signed consent). The interface, user access etc for the program worked perfectly fine without issue.

Referee #2

(Remarks to the Author)

Review of "Towards Accurate Differential Diagnosis with Large Language Models"

This is my brief review of the revised version of this paper. I am impressed by the care with which the authors have responded to both critiques from the initial round. Although I very much appreciate the opportunity to access the system as part of the review, I have not done so just because of a lack of time. I am already feeling guilty about the delay in getting to this 2nd review and don't want to delay further.

I do have a few stylistic suggestions, but would not need to review any changes these lead to.

1. Figure 4 uses colors to differentiate responses to the rating questions, but gray scale in the x-axis labels. It would help to use the same colors there as well.
2. When describing the different LLMs used in the study (Med-PALM 2, AMIE), it would help the reader to mention some statistics for each, such as the total number of parameters of the model, or the total computational effort including the size of the pre-training data and fine-tuning data or perhaps FLOPS used in training them.
3. There are minor bugs in the references. E.g., the journal is missing in 34 (BMJ, I think), "Hello AI" has non-curly quotes and an extra space in 35, "Anthony Celi, L." should be "Celi, L. A." in 15. I didn't check all of these carefully, but someone should.

I am satisfied that the paper is now ready for publication and have no further criticism.

(Remarks on code availability)

Google LLC
1600 Amphitheatre Parkway
Mountain View, CA 94043

650-253-0000 main
google.com

Oct 31st 2024

Dear Editors and Reviewers,

We would like to thank you for your thoughtful, thorough, and constructive feedback. We are glad you appreciated the analysis in our manuscript and found the article to be clear and concise. We believe that this work is an important step toward validating the efficacy of large language models (LLMs) in supporting differential diagnosis, a foundational task within clinical medicine. Below we respond to each of the comments and summarize the changes we have made to the manuscript. We have included a marked-up version of the manuscript highlighting changes corresponding to comments. To help make the changes clear we have made edits corresponding to comment from Referee #1 **in red** and Referee #2 **in green**. However, we encourage both reviewers to read both sets of edits as some similar comments were made by both reviewers. Thank you again for the thoughtful reviews which have led to material improvements in the manuscript.

Before we begin we would like to highlight that **we are able to provide access to the model** via the interface we used for our experiments (DataCompute). Access instructions are provided in Appendix A. A login 1) username and 2) password will be provided separately after the trusted tester agreement has been signed and returned - we will coordinate with the Editor to facilitate this. The trusted tester agreement is included as a Related Manuscript File provided with the submission. We thank you for your patience while we set this interface up and for accommodating the extra step of completing the testing agreement.

Response to Referee #1

Comment: *The article explores the performance of AMIE in differential diagnosis of 302 NEJM Clinicopathological conferences with comparison to 20 board-certified internists. The internists all perform the differentials unaided and are afterwards allowed aid by either search only or AMIE (and search). Specialists in the appropriate fields review the results based on Top-n-accuracy, comprehensiveness of DDx lists, quality of the lists and appropriateness. The article finds that unaided, the internists are outperformed on every metric. Clinicians with assistance outperform unaided clinicians, with clinicians aided by AMIE performing the best. AMIE alone outperforms all clinician groups. I find the article to be original and of great significance. Whilst as the article states itself, LLMs have already been shown to perform well*

Google LLC
1600 Amphitheatre Parkway
Mountain View, CA 94043

650-253-0000 main
google.com

on NEJM cases, the performance of LLMs has yet to be compared to verified doctors. This article presents compelling evidence that LLMs, at least AMIE, can indeed outperform board-certified internists. Regarding clarity and context, I found the entire article, including abstract, to be clear, concise and appropriate. I particularly enjoyed the discussion. I want to extend my congratulations to the authors for a tremendous work, I was happy to be allowed to review it, and I want to assure the authors that I believe this a very high-quality article with major impact, despite my comments above. I want to apologize for the length, but it feels like an important article and an important journal. I want to thank Nature for the chance to review this article, it has been an honor. Regarding conclusions. The overall conclusion that AMIE outperforms unassisted doctors seems very well supported. The conclusion that aided doctors (both groups) outperforms unassisted doctors also seems well supported by the article. Regarding methods: Overall, I appreciate the methods of the article. From a technical standpoint they seem valid overall, and the authors has generated and presented a large dataset. I appreciate the contamination analysis as well, as I imagine this would otherwise be a point of contention.

Response: We appreciate the reviewer's positive and thoughtful comments and suggestions. We are glad that the methods we used were appreciated and that you felt that the conclusions were well supported.

Access to the Models

Comment: *1. I choose the word 'seem' as I do not have access to AMIE (nor even Med-PaLM2). Access requests have been futile. I cannot verify the data the article presents; however, I can say that with my experience with other LLM in a similar dataset with similar prompts, the data does not at all seem unrealistic, especially for an LLM tuned in medicine.*

Response: Thank you for your comment. We have uploaded the full set of DDx lists from the physicians and the models. We also plan to release the full set of DDx lists publicly to accompany the publication. The NEJM cases themselves are publicly available and can be obtained from the publisher; however, we do not have license to distribute the copyrighted text to third parties ourselves. For licensed access please contact Robert McKinney (rmckinney@nejm.org) from the New England Journal of Medicine. If the editor would be amenable to coordinate to avoid compromising review anonymity we would be very grateful. **The DDx lists can be found in supplementary material submitted with this manuscript.**

Google LLC
1600 Amphitheatre Parkway
Mountain View, CA 94043

650-253-0000 main
google.com

We are able to provide access to the model via the interface we used for our experiments (DataCompute). An appendix (**Appendix A**) is provided at the end of this document with instructions for access to the interface. In summary, the URL is:

https://datacompute.google.com/w/paper_reviewer

A login 1) username and 2) password will be provided separately after the trusted tester agreement has been signed and returned - we will coordinate with the Editor to facilitate this. The trusted tester agreement is included as a Related Manuscript File provided with the submission. We thank you for your patience while we set this interface up and for accommodating the extra step of completing the testing agreement.

The model provided via the DataCompute interface is the exact model used for our experiments. In addition, the base model used for our work is the same as the base model used for the MedLM model which is available via Google Cloud.

Consistency

Comment: *2. I would have liked to see some data on consistency – the article would benefit from analysis on repeated runs.*

Response: Thank you for this comment, we appreciate the request for data on the consistency of the results. We have added two additional sets of results related to 1) the consistency of model outputs and 2) the consistency among distinct groups of raters respectively.

1. **Consistency of the model outputs.** LLMs can be non-deterministic if they have a temperature that is non-zero or if the batch size is changed. We generated 10 repetitions of the DDx lists from AMIE for all 302 cases with temperature=0.5 (Fig. R1(a)). We then performed auto evaluation on these using Med-PaLM 2. We also performed experiments with temperature settings of 0, 0.2, 0.4, 0.6, 0.8, 1.0 (Fig. R1(b)). These results show that the model produces consistent results. Better performance is generally obtained with a non-zero temperature, particularly at higher “n”. These results have been added to Appendix A (Model Consistency).

(a) AMIE Top-n performance, based on auto-eval using Med-PaLM 2, repeated 10 times. Temperature=0.5.

(b) AMIE Top-n performance, based on auto-eval using Med-PaLM 2, with different temperature settings.

Figure R1. Consistency Experiments.

2. **Consistency between raters for the DDx generation without assistance.** Comparing the two conditions, the average scores for appropriateness and comprehensiveness of the DDx lists *without* assistance were slightly different; however, performing Wilcoxon signed-rank tests the appropriateness and comprehensiveness scores were not statistically different suggesting consistency among distinct groups of raters under the same unassisted circumstances:

Appropriateness:

Search Baseline = 3.71

AMIE Baseline = 3.75

Wilcoxon signed-rank test $p = 0.63$

Comprehensiveness:

The distribution of comprehensiveness scores were similar:

The DDx contains all candidates that are reasonable: 70 vs 82

The DDx contains most of the candidates but some are missing: 88 vs 83

The DDx contains some of the candidates but a number are missing: 105 vs 89

The DDx has major candidates missing: 39 vs 48

Google LLC
 1600 Amphitheatre Parkway
 Mountain View, CA 94043

650-253-0000 main
 google.com

The number of cases that scored 4 (i.e., The DDx contains all candidates that are reasonable) was not statistically different for clinicians in the baseline Search (condition I) and AMIE (II) conditions.

McNemar’s Test: p = 0.23

Quality:

The quality scores were slightly different:

Quality Score:

The number of cases that scored 5 (i.e., The DDx included the top diagnosis) was higher in the Search condition baseline (33.8%) compared to the AMIE condition baseline (27.2%).

McNemar’s Test: p = 0.03

These results, along with trends from Figure R3 above (Figure 5 in the manuscript), suggest that baseline distributions were statistically similar with respect to several endpoints, and that for those endpoints where differences could be determined, clinicians in the AMIE condition started off from a slightly lower baseline. The latter observation, if anything, may further corroborate (rather than undermine) our main conclusion that AMIE had a stronger assistive effect than Search alone because results after assistance from AMIE were greater than those after assistance with search.

To further consolidate this point we ran a linear mixed effects models to test the effect of the Arm (either Assisted by Search=0 or Assisted by AMIE=1) on the final diagnosis score after assistance while controlling for the effect of baseline (unassisted final diagnosis score).

$$\text{Assisted Final Diagnosis Score} \sim \text{Arm} + \text{Unassisted Final Diagnosis Score} + (1|\text{Clinician}) + \epsilon$$

The results were as follows:

No. Observations: 604, Converged: Yes

	Coefficient	Std. Err.	z	P> z	CI: 0.025	CI 0.975
Intercept	1.438	0.120	12.028	0.000	1.203	1.672
Arm	0.378	0.085	4.451	0.000	0.212	0.545

Google LLC
1600 Amphitheatre Parkway
Mountain View, CA 94043

650-253-0000 main
google.com

Baseline	0.620	0.030	20.338	0.000	0.560	0.680
Group Var	0.001					

The model confirms that the Arm has a significant effect on the final diagnosis quality score when controlling for the baseline.

These results have been added to Appendix B (Rater Consistency).

Number of Diagnoses

Comment: *3. I wonder about the “top 10 diagnoses” approach. Considering 10 diagnoses would generally be a lot in real life and I have doubts about the clinical relevance of this. I don’t know if there are any data on this question, but this very article finds that the median is 6 diagnoses for unassisted doctors. This warrants discussion in my opinion.*

Response: We appreciate this suggestion. There are two results we would like to highlight that we believe help elucidate whether the length of the lists impacted the results.

(1) The result for six diagnoses (N=6) for each case. The median list length was 6 and the top-N accuracy of AMIE for N=6 was 51%.

(2) We also performed a new variable top-N experiment, where for each case N was set to the number of diagnoses the human clinician had generated for that case, thereby capping AMIE’s DDx list to the same length as the corresponding DDx list from the human clinician for each case. The variable top-N performance of AMIE was 59.4%, which is similar to the performance at N=9 and 10.

In addition, while we agree that 10 diagnoses is generally “a lot in real life”, it is also worth highlighting that the cases (as the reviewer notes) are tough diagnostic challenges, with the gold-standard DDx lists provided in the case discussions themselves having at least 5 diagnoses, with many greater than 10. Our gold-standard DDx lists had a minimum of 5 diagnoses and a median of 7.

We have added these results to Section 6.1.

Google LLC
1600 Amphitheatre Parkway
Mountain View, CA 94043

650-253-0000 main
google.com

Comment: 4. *In relation to this, I am concerned about the blinding of the human raters. If the doctors output a mean 6 diagnoses (unaided) and AMIE always outputs 10 diagnoses, it seems there is a risk of being able to figure out which is which. The article makes no mention of how the doctors were instructed to write their diagnoses, so spelling mistakes, abbreviations and differences in connotation could exacerbate this issue. This seems to be a potential point of bias and any potential conflicts of interest. Elaboration would improve the article.*

a. In relation to this, how were the participating doctors (diagnosers and evaluators) incentivized? I think the article should state this for transparency.

Response: We appreciate this thoughtful consideration regarding rater blinding and incentives. While we were careful to make sure that the casing and spellings of the clinicians' responses were not incorrect or obviously different compared to the AMIE-generated DDx lists, we do acknowledge that there were some systematic differences in numbers between the expert and AI DDx lists. However, we believe that this did not impact our results for several reasons:

1. Due to a similar concern during the execution of our study, we quality control checked the lists from both the model and the clinicians before running the evaluation ("Step 2") to ensure that there were no obvious formatting differences. We were careful not to change the content of their lists, but just the formatting. The reviewer can check the DDx lists provided in the supplemental material uploaded through the portal.
2. The raters *did not know ahead of time* the various potential sources of DDx lists and that these could include AI models. During the rating process, the ordering of DDx lists was randomized and DDx lists were labeled as stemming from "Expert A", "Expert B", "Expert C", "Expert D" or "Expert E". We would emphasize that neither the raters nor us (as study designers) knew ahead of time what the length of the DDx lists would be for the clinicians. The clinicians in Step 1 were different from the clinicians in Step 2 and therefore those evaluating the DDx lists (clinicians in Step 2) did not know the number of diagnoses those generating the candidate lists (clinicians in Step 1) were asked for. In principle they should not have been able to guess which source the lists came from.
3. Finally, the trends from human and auto evaluation were similar. Autoeval was definitely "blinded" to the source of the data.

Nevertheless, we appreciate this comment and have added text to the Limitations section that this is a concern in principle.

Google LLC
1600 Amphitheatre Parkway
Mountain View, CA 94043

650-253-0000 main
google.com

Regarding incentives, participating doctors (diagnosers and evaluators) were recruited through vendor companies based on requested medical specialties (e.g., internal medicine or subspecialties) and credentials. The clinicians were remunerated by the vendor companies at hourly market rates based on specialty. We have added clarification under a new Methods section ‘Clinician Incentives’.

Comparison with and without Labs

Comment: 5. *The article compares a group that has access to labs and one that does not. I am aware of model limitations, but doctors using the model in real life would probably be tempted to provide labs in the prompt, meaning it may not be a realistic comparison either. GPT-4 seems to perform adequately with labs. If possible, I believe it should be added for a more complete comparison, but this might not be feasible.*

Response: We appreciate the suggestion to include lab values in the prompts. Of note, because the case texts represent a synthesis of available clinical data, case texts do routinely include a subset of key lab values—though not all lab values. Indeed, there are cases that include tables of lab values that we did not include in our prompts initially and which the physicians had access to. This is a great suggestion and we have run an additional set of experiments in which we have included the tabular lab data in the prompt, run our automated evaluation and provided a comparison. For the 302 cases, 239 had specific tables of “Laboratory Data”. We included the contents of these as tab delimited text in the query with the case text. We performed automated evaluation on 20 repetitions of the DDx task with Temperature=0.5. The Top-N results (see Fig. R2 below) were statistically similar to the results without labs included in the input, although there is a slight indication that they contain information above and beyond that in case text that the model was able to leverage.

Figure R2. Top-n accuracy including Lab Data with Case Presentation Text. Comparison of the top-n accuracy scores based on auto-evaluation when using the case text (original experiments) and case text + laboratory data table as delimited text.

These results have been added to Appendix C.

We have also added discussion to the manuscript (Limitations section) about the growing number of multimodal models which could be evaluated in a similar way.

Statistical Tests

Comment: 6. Page 8: “The mean appropriateness score of the LLM (4.34)[...]” - The data most likely does not follow a normal distribution (as they can only be a few whole integers), so he would recommend a signrank test.

7. Page 9: “The mean appropriateness score after assistance [...]” – Same as above.

These are my own comments regarding statistics:

8. Did the contamination analysis change anything from significant to insignificant? It is vaguely described, only specifically mentioning significance for clinicians assisted by LLM vs without LLM.

Response: Thank you for raising these points with respect to statistical testing. We agree it is helpful to have additional statistics given the skew in distribution of appropriateness scores.

Google LLC
1600 Amphitheatre Parkway
Mountain View, CA 94043

650-253-0000 main
google.com

Following your suggestion, we conducted Wilcoxon signed-rank tests which supported our conclusions.

The mean appropriateness score of AMIE (4.34) was significantly higher than that for unassisted clinicians (3.74):

Paired t-test $p < 0.001$

Wilcoxon signed-rank test $p < 0.001$

The mean appropriateness score of AMIE (4.34) was significantly higher than that for the assisted clinicians in the Search (3.80) condition:

Paired t-test $p < 0.001$

Wilcoxon signed-rank test $p < 0.001$

The mean appropriateness score of AMIE (4.34) was significantly higher than that for the assisted clinicians in the AMIE (4.06) condition:

Paired t-test $p < 0.001$

Wilcoxon signed-rank test $p < 0.001$

These results have been added to Section 6.1.

With regard to the contamination analysis, on the non-overlapping dataset (cases from 2022 onwards) the DDx ratings mostly remained consistent despite the much smaller sample ($N=56$). Specifically, the DDx quality scores remained significantly higher for clinicians assisted by our LLM (top-10 accuracy 52.3%) compared to clinicians without its assistance (34.6%) (McNemar's Test: $p < 0.01$) on the uncontaminated set. The quality score for the LLM (55.4%) was also higher than that for the clinicians without assistance (McNemar's Test: $p < 0.01$). The main difference was between clinicians after assistance in the search (46.2%) and AMIE conditions (52.3%) which was no longer significant (McNemar's Test: $p = 0.39$). This final result is likely a consequence of the reduced sample size ($N=56$).

On the additional overlap analysis excluding only articles with greater than 512- characters overlap all the differences that were significant on the full set remain so. The DDx quality score remained significantly higher for clinicians assisted by our LLM (top-10 accuracy 51.4%) compared to clinicians without its assistance (33.1%) (McNemar's Test: $p < 0.01$) on the uncontaminated set. The quality score for the LLM (61.4%) was also higher than that for the

Google LLC
1600 Amphitheatre Parkway
Mountain View, CA 94043

650-253-0000 main
google.com

clinicians without assistance (McNemar's Test: $p < 0.01$) and the difference between clinicians after assistance in the search (44.2%) and AMIE conditions (52.4%) was significant (McNemar's Test: $p = 0.05$).

These results and clarifications have been added to Section 6.5.

Qualitative Analysis:

Comment: *9. I would have liked at least a bit of data on the qualitative analysis, I think this is a very interesting aspect, but perhaps it extends outside the scope of this article.*

Response: We are glad you appreciated our mixed-methods approach. Following your guidance, we have expanded on our qualitative analysis in Methods, Results and Discussion as follows:

Methods - 5.2 Qualitative Interviews:

"We conducted a thematic analysis of notes from interviews taken by researchers during the interviews, employing an inductive approach to identify patterns (themes) within the data. Initial codes were generated through a line-by-line review of the notes, with attention paid to both semantic content and latent meaning. Codes were then grouped based on conceptual similarity, and refined iteratively. To enhance the trustworthiness of the analysis, peer debriefing was conducted within the team of researchers. Through discussion and consensus, the final themes were agreed upon."

Results - 6.7 Qualitative Analysis:

"Qualitative analysis of semi-structured interviews with clinicians revealed the following themes:

- **LLM for Generating Comprehensive Differentials:** *The LLM was particularly effective at broadening the scope of differential diagnoses, prompting clinicians to consider diagnoses they might not have initially considered, particularly in less familiar medical specialties (C3, C5). They noted that this was helpful when their initial differential was limited, or when they were uncertain about the diagnosis and struggled to identify useful search terms. This insight aligns with C2's observation that search tools were less helpful when they were unsure of the potential diagnoses.*

Google LLC
1600 Amphitheatre Parkway
Mountain View, CA 94043

650-253-0000 main
google.com

- **Need for Clinical Judgment and Critical Thinking:** While the LLM could expand the diagnostic considerations, clinicians consistently highlighted the model's limitations and the need to apply clinical judgment. They noted that the LLM could make mistakes, particularly in complex or atypical cases (C5) and could be influenced by the line of questioning, which could be misleading for individuals without medical expertise (C1, C3). This point is consistent with C2's view that the LLM was intuitive for them as a clinician but might be inaccurate for those without a clinical background.
- **Variable Effectiveness Based on Case Complexity:** The LLM's helpfulness varied depending on the specifics of the case presentation. It was perceived as most useful for straightforward cases with clear pathognomonic signs, offering a quick and efficient way to confirm or expand upon the clinician's initial impressions (C1, C3). However, the model's limitations were more apparent in complex cases. Clinicians noted it could become distracted by individual symptoms and fail to synthesize the information holistically, which limited its usefulness in these scenarios (C1).
- **LLM as a Collaborative Learning Tool:** Clinicians viewed the LLM as a valuable learning resource, particularly for medical education, due to its ability to quickly provide a range of potential diagnoses and their explanations from a single source (C1, C3). They highlighted the potential for the LLM to 'upskill' clinical providers through this expanded perspective and the provision of detailed explanations (C1, C3). Clinician C2 expressed a desire to have this tool available every day as an aid for their clinical practice.
- **Interface and User Experience:** Clinicians found the conversational interface intuitive and easy to use (C2), highlighting that this enabled them to interact with the LLM efficiently without adding substantial time to the task. However, they indicated the LLM's helpfulness could be improved by providing more explicit guidance on how to interact with it effectively (C5) and by enabling the model to communicate uncertainty."

Discussion (Section 7):

*"Our qualitative findings from semi-structured interviews with clinicians highlight the collaborative nature of the diagnostic reasoning process and the importance of clinical judgment when using an LLM. While the LLM was capable of generating a broad differential diagnosis in isolation, the clinicians' expertise enabled them to filter these suggestions when they were using the tool, discarding those deemed inaccurate or irrelevant and leading to a more conservative final differential list. This active evaluation and filtering process could explain the gap between standalone LLM performance and clinician performance when assisted by the tool, with several specific factors highlighted: (1) **Anchoring Bias:** Clinicians tended to anchor on their initial, unassisted DDx. This is consistent with known anchoring biases and might be exacerbated by the two-stage study design; (2) **LLM Suggestibility:** Several clinicians noted that the LLM could*

Google LLC
1600 Amphitheatre Parkway
Mountain View, CA 94043

650-253-0000 main
google.com

*be led down alternative diagnostic paths by their follow-up questions and that this could lead to inaccurate conclusions that clinicians recognized as not being supported by the evidence; (3) **Trust Calibration:** Clinicians highlighted the importance of the model being able to communicate when it is unsure, as this would likely have impacted the extent to which they trusted and incorporated the LLM's suggestions.”*

Comment: 10. *The article concludes that doctors using AMIE outperformed doctors using search on top-n-diagnoses. I would like to challenge the extent to which it is valid based on the clinical relevancy of this conclusion. The difference does seem arise for sure (only graph shown) until top 5 diagnoses by human evaluation or top 6 diagnoses by automated evaluation. The article reports that the unaided doctor puts a median of 6 diagnoses in their DDX in these case challenge scenarios.*

Response: We thank the reviewer for this fair comment. As described above, we have added experimental results with variable length AMIE lists to provide confidence that the results are still consistent when the model provides the same number of differentials as the human clinicians both on average and on a per-case level. See Section 6.1.

Human vs. Auto Eval:

Comment: 11 & 12. *Regarding human vs automatic evaluation, the article concludes that there is reasonable agreement between the two. While the two graphs do look somewhat similar, the data is insufficient to support the conclusion (no supplementary either).*

a. I ran this bullet by our statistician as well, who also reckoned the statement is too vague for the shown data. This conclusion is later used to build the basis for concluding that AMIE outperforms GPT-4 (based on automatic evaluation). However, based on the aforementioned lack of data supporting this, I believe this conclusion is put into question. This conclusion is also put into question by the contrast to the results of GPT-4 in Kanjee et al. rated by humans.

Response: We agree that this statement warrants corroboration, and have added statistical tests to more robustly compare the human and automated evaluation. In particular, we computed Cohen's kappa as a measure of agreement between human raters and automated evaluation with respect to the binary indicator of whether a given diagnosis, i.e. an individual item from a proposed DDX list, matched the correct final diagnosis. Cohen's kappa for this matching task was 0.631 indicating “substantial agreement” between human raters and our

Google LLC
1600 Amphitheatre Parkway
Mountain View, CA 94043

650-253-0000 main
google.com

automated evaluation method per established guidelines from Landis and Koch (1977). This result has been added to Section 5.1.

We do not believe that our results contradict the numerical results in Kanjee et al. (2023). In that work there was no comparison to AMIE or to human clinicians. The reason the numbers on the subset of 70 cases may be slightly different might be the somewhat subjective nature of assessing whether a DDX list contains the correct diagnosis or not. Another possible contributing factor could be the evolving nature of ChatGPT over time as documented by Chen et al. (2023).

As one illustrative example, in:

Case 7-2021: A 19-Year-Old Man with Shock, Multiple Organ Failure, and Rash

The final diagnosis was: Meningococcal Purpura Fulminans

The AMIE top diagnosis was: Sepsis

The AMIE second diagnosis was: Meningococemia

The AMIE diagnosis of Sepsis is technically not wrong; however, it is not very actionable due its broadness. Whereas Meningococemia is better and Meningococcal Purpura Fulminans is quite specific and actionable as a possible diagnosis.

While the rubric for evaluating whether a diagnosis is included in a DDX list is clear, it is possible to disagree whether an individual diagnosis is specific enough to be counted as correct vs. incorrect. We have emphasized these points in the revised manuscript, including elaboration regarding nuances that need to be considered for comparing our results to those presented by Kanjee et al.; we believe, findings from Kanjee et al. support and complement the findings in the prior work. We have added these points to the Limitations section.

Time Analysis

Comment: 13. *The “time spent” analysis is in my opinion not rooted in reality and I question the validity, as it neglects the fact that the presented case was pre-populated, something a doctor using an LLM would have to write themselves from the EMRs. Despite being a proponent of AI I use search a lot more than LLM in my clinical practice due to this time factor alone.*

Google LLC
1600 Amphitheatre Parkway
Mountain View, CA 94043

650-253-0000 main
google.com

Response: We acknowledge the reviewer’s consideration that the analysis of time spent on the tasks does not map to time taken in real settings. We appreciate that, in real settings, a clinician would need to write a case description/notes. We also recognize that the way an LLM is used versus Internet search is different and the time comparison does not account for this. We have added these important considerations to the Limitations section.

Comment: *The article mentions supplementary data which is missing, as far as I can tell.*

- I stumbled upon a few incomplete sentences missing words, which should not be present in a high-profile article such as this. Example page 8 line 5 of main text (following “The mean appropriateness [...]”) and page 11 line 6 of main text (sentence: “. performs better”). A read-through is in order.

- The article does not mention specifically how many specialists evaluate each case. I assume 1.

- I believe figure 3 and the text does not match completely? AMIE DDx is put as “Condition II”, but in condition II the article describes that the AMIE DDx is only a suggested first prompt, suggesting that the doctors in condition II could put something else (e.g. top 5 or something unrelated). The LLM was assessed on its own, not as part of condition II, correct?

- Might be a language barrier issue, but “The DDx contains all candidates that are reasonable” is confusing to me, I hope it can be clarified a bit. Based on the category 2 and 3 title it seems that it would mean that all the reasonable diagnoses from the Gold DDx list is included? But why would the Gold DDx list include unreasonable diagnoses?

I found the references appropriate, but “Towards Conversational Diagnostic AI” by Tao Tu et al. was not mentioned despite exploring the same model and by some of the same authors. (Was released on Arxiv after this one)

Response: We appreciate these comments and address each below.

We have reviewed the manuscript and corrected incomplete sentences, including the two pointed out in this comment.

There is a level of subjectivity to whether DDx entries are reasonable or not. While the Gold DDx lists were created by specialists, another clinician may be of the opinion that another candidate should have been included in the list. Therefore, we asked whether the DDx list contained **all** the candidates that were reasonable.

Google LLC
1600 Amphitheatre Parkway
Mountain View, CA 94043

650-253-0000 main
google.com

Nineteen (19) U.S. board-certified specialist clinicians were involved in stage 2 of the study and each case was evaluated by one of these clinicians. Due to the time consuming nature of creating the DDx lists and evaluating the five candidate lists for 302 cases, it was not possible to have multiple specialists evaluate each case. We estimate that the experiment we conducted required 200 hours of clinician time.

Indeed the work “Towards Conversational Diagnostic AI” was published after this paper. However, it does use the same model and we have reference to it in this paper.

We would like to clarify Fig.3 and its caption. The LLM was assessed on its own (AMIE DDx) AND in the condition where humans were assisted by AMIE (Condition II). In Condition II the *first* prompt to AMIE was suggested to be: “What are the top 10 most likely diagnoses and why (be precise)?”; however, as you correctly note, the clinicians could put something else (e.g. top 5). We have clarified this in the caption in the revised manuscript.

Thank you for noting that the appendix (supplementary material) with the list of case IDs was not included in the PDF. We have added those to Appendix C, the list of case IDs and article titles can also be found in the spreadsheet with the DDx lists which has been submitted as supplementary material alongside this revision in the Nature portal.

Comment: *Somewhat; As mentioned in reviewer comments, AMIE and Med-PaLM2 are not accessible to me. Thus, I cannot test the results of the prompts in the actual API. However, I've done very extensive testing of similar prompts and also tested their prompt in GPT-4. Whilst GPT-4 is not the same and LLMs are not interchangeable (even different versions of the same LLM might not give the same response), the prompts SHOULD work in an advanced LLM trained in medicine. I've used the publicly available version of Gemini which functions at a high level, so I have no reason to believe Google would not be able to tune an LLM to medicine. In so many words, based on my experience with LLMs this would work, and I believe it would work with other advanced LLMs as well. HOWEVER, if you want a guarantee that AMIE can actually answer what they say it can (e.g. test the prompts in the actual program), you will have to pressure Google into providing access. I think your name carries a lot more sway than mine in this regard :-)* I've previously requested access from Google to their LLMs without luck and I have done so again due to this review, although I've not told them I wanted it for this review specifically as I did not want to break any confidentiality rules, but maybe that would help.

Google LLC
1600 Amphitheatre Parkway
Mountain View, CA 94043

650-253-0000 main
google.com

Response: We appreciate this comment and apologize for the frustration in getting access to test the medically tuned models. **We are able to provide access to the model** via the interface we used for our experiments (DataCompute). An appendix (**Appendix A**) is provided at the end of this document with instructions for access to the interface. In summary, the URL is:

https://datacompute.google.com/w/paper_reviewer

A login 1) username and 2) password will be provided separately after the trusted tester agreement has been signed and returned - we will coordinate with the Editor to facilitate this. The trusted tester agreement is included as a Related Manuscript File provided with the submission. We thank you for your patience while we set this interface up and for accommodating the extra step of completing the testing agreement.

The model provided via the DataCompute interface is the exact model used for our experiments. In addition, the base model used for our work is the same as the base model used for the MedLM model which is available via Google Cloud.

Response to Referee #2

Comment: *The paper examines the performance of “AMIE, an LLM model optimized for differential diagnosis, alongside a user interface allowing clinicians to interact with the model for improving clinical diagnostic reasoning” on the task of creating differential diagnosis lists from 302 NEJM CPC cases published in the past decade. AMIE is based on the Large PaLM 2 model, fine-tuned with a variety of available medical QA datasets, which were also used in this group’s 2023 paper on medical question answering, plus apparently some others not described (“The datasets used included ...”).*

The headline result is that after a panel of domain experts determined the gold standard diagnoses for the CPC cases, AMIE on its own included the correct diagnosis among its top 10 generated ones in 59.1% of the cases and nailed the top diagnosis in 29%. Human experiments with 20 board-certified physicians with a median of 9 years’ experience showed that AMIE’s performance on the top-10 task was significantly better than the doctors’, whose top-10 accuracy was only 36.1%, or 44.4% when encouraged to use Web search. Oddly, their performance when using AMIE, whose goal is to improve their abilities, was 51.7%, considerably lower than what AMIE achieved by itself, though much better than their unassisted performance.

Google LLC
1600 Amphitheatre Parkway
Mountain View, CA 94043

650-253-0000 main
google.com

Response: We appreciate the reviewer’s thorough comments and would like to clarify details about the datasets used in our work. Below is a summary of the fine-tuning datasets with more details:

Medical Reasoning. 11,450 questions from the MedQA consisting of US Medical Licensing Examination multiple-choice style open domain questions were included in the training set and 1,273 questions in the test set. We curated 191 MedQA questions from the training set where clinical experts crafted step-by-step reasoning leading to the correct answer.

Long-form Medical Question Answering. For long-form MedQA examples we used expert-crafted long-form responses to 64 questions from HealthSearchQA, LiveQA, and Medication QA in MultiMedQA.

Real-world Dialogue. A dialogue dataset of 218 dialogues between a clinician and patient and their associated ground truth diagnoses was also incorporated into the training set. The dataset was heavily skewed towards respiratory (n=168) and musculoskeletal (n=40) dialogues, but also contained cardiology (n=5), gastroenterology (n=4), and dermatology (n=1) texts. In total, 102 unique diagnoses are represented across these dialogues, and the mean number of turns was 98.3 (25th percentile=85.0, 75th percentile=113.0).

We have added similar content to the revised manuscript in Section 3 (“Training a Large Language Model for DDx”).

Main Conclusion

Comment: *One of my concerns with this paper is that these results suggest a take-away lesson that we should supplant physicians by AMIE-like systems for clinical diagnosis: “the proposed LLM outperformed an unassisted board-certified physician in both top-1 and top-n performance”. To be fair, the authors do not suggest this, but I suspect that many readers will settle on that conclusion, driven by the reported results. I wish the paper included a serious attempt to understand why there is this gap. Previous claims for earlier physician assistance systems have often argued that the combination of a physician and a computational model do outperform either one by itself, yet this paper’s results do not conform to that expectation. The diagram in Figure 6 summarizes the aggregate effect of adding search or LLM, but it would be extremely helpful to investigate what actually changes in the doctors’ minds as they incorporate this additional information. What sorts of cases are helped by these decision support tools and*

Google LLC
1600 Amphitheatre Parkway
Mountain View, CA 94043

650-253-0000 main
google.com

what are harmed? How does the difficulty of the case (however that is judged) affect this? I'd be surprised if it were just random.

Response: We agree that the conclusion from reading our paper should not be to supplant physicians by AMIE-like systems for clinical diagnosis. We conducted qualitative analysis as part of our study to help inform the answer to why there was a gap between the model and the clinician aided by the model. We added further elaboration on qualitative analysis in Methods, Results and Discussion per request from Referee #1 (see above). Regarding Referee #2's question on the gap between standalone LLM performance and clinician performance when assisted by the model, we expanded our Discussion section as follows:

*“Our qualitative findings from semi-structured interviews with clinicians highlight the collaborative nature of the diagnostic reasoning process and the importance of clinical judgment when using an LLM. While the LLM was capable of generating a broad differential diagnosis in isolation, the clinicians’ expertise enabled them to filter these suggestions when they were using the tool, discarding those deemed inaccurate or irrelevant and leading to a more encompassing and considered final differential list. This active evaluation and filtering process could explain the gap between standalone LLM performance and clinician performance when assisted by the tool, with several specific factors highlighted: (1) **Anchoring Bias:** Clinicians tended to anchor on their initial, unassisted DDX. This is consistent with known anchoring biases and might be exacerbated by the two-stage study design; (2) **LLM Suggestibility:** Several clinicians noted that the LLM could be led down alternative diagnostic paths by their follow-up questions and that this could lead to inaccurate conclusions that clinicians recognized as not being supported by the evidence; (3) **Trust Calibration:** Clinicians highlighted the importance of the model being able to communicate when it is unsure, as this would likely have impacted the extent to which they trusted and incorporated the LLM’s suggestions.”*

Comment: *One speculation I can offer for this surprising result is that the experiments they ran included two conditions for generating the DDX list, the first using no reference tools, followed by encouragement to use internet search or “other resources” (?), followed by use of the LLM, search, etc. From what we know about “anchoring” phenomena in human cognition, perhaps this ordering allows the doctors to fixate on their initial opinion and to show reluctance to accept apparently better advice fully. Unfortunately, it would probably take a much larger sample of subjects to show significant results among the conditions if the experiments exposed each subject to just one actual condition (review of HPI, review with search, review with LLM).*

Google LLC
1600 Amphitheatre Parkway
Mountain View, CA 94043

650-253-0000 main
google.com

Response: We thank the reviewer for encouraging us to consider potential “anchoring” effects which we have incorporated in our Discussion of qualitative insights (see above) and provide more detail on below. To avoid potential reader confusion regarding the study procedure, we have clarified the terminology in the manuscript in that the clinicians were only assigned to a single condition per case (unassisted and then search, or unassisted and then LLM). Specifically, they created an initial DDx list with no reference tools, and then EITHER (1) a list with the aid of Internet search OR (2) AMIE. We appreciate the reviewer’s considerations of potential anchoring effects to their INITIAL list, which we considered in the study design. We believe that in our study design anchoring effects described by the reviewer should impact BOTH conditions equally since in both the clinicians were asked to update their DDx lists once and only once. For further consideration on the trade-off between sequential and independent designs and potential bias effects, we would like to point to an FDA guidance on reader studies (‘Joint FDA-MIPS Workshop on Methods for the Evaluation of Imaging and Computer-Assist Devices’ ; PMC5557046): “The workshop speakers agreed that the sequential study design was acceptable for three basic reasons beyond representing intended use. *First, the literature has several examples comparing the results of sequential and independent study designs, none of which shows a significant bias (ie, a bias that changes the overall study results) (73–76)*. Second, the sequential design builds in a correlation between the unaided and aided performance, which improves the power over the independent design to detect a difference in performance. Third, the sequential design is less burdensome compared to the independent design on readers and reading time because all the data is collected in one session.”

Furthermore, we also considered the potential for clinicians to “over rely” on the AI. There is evidence of similar reader studies where the model performance tends to “pull” readers towards the AI performance which provides evidence against strong anchoring biases to some extent. We have added limitations to this effect in the paper (Section 8). We appreciate the reviewer’s thoughtful points, which we agree are important to acknowledge in the manuscript.

Comment: *The superior performance of the LLM is especially surprising since AMIE was given only the HPI of the CPC cases to work from, whereas the physicians also received admission labs and imaging studies. Because these are often sources of decisive information, I would have thought the physician performance might be better, given their access to more data. The paper says that such data were “sometimes included in the case description”—a rather vague explanation. Without some further insight into this question, I am left with the suspicion that the authors (and I) are missing something important.*

Response: We thank the reviewer for this nuanced consideration. The way that the NEJM cases are written, as a *synthesis* of a complex case presentation, leads to important data from Tables being referenced in the text—very similar, though not exactly, to how physicians summarize case presentations to other care providers and selectively include the most pertinent information for the case (including key lab values and imaging findings). Put differently, the included lab values or imaging findings in the case text presented to the model are not random but deemed by the expert to be highly relevant to the case. To make this point quantitative we have reviewed all 302 cases and added metadata to reflect that N=239 had tables of labs that were referenced in the text. In our original experiments, AMIE (model) only had access to the main body of the case text and not the Tables. Based on excellent suggestions from both Referees #1 and #2, we have conducted additional experiments inserting lab values from Tables into the AMIE prompt; as we note above, our results remained unchanged.

Figure R2. Top-n accuracy including Lab Data with Case Presentation Text. Comparison of the top-n accuracy scores based on auto-evaluation when using the case text (original experiments) and case text + laboratory data table as delimited text.

Comment: *Performance evaluation in the paper is based not only on top-n accuracy but also by asking experts to provide Likert scale-like evaluations of quality, comprehensiveness, and appropriateness. Although such evaluations can tease out more subtle criteria, such as the clinical importance of diagnoses that may have been missed or inappropriately included, my experience makes me suspicious of such “did you like it” methods. For example, Figure 4 shows*

Google LLC
1600 Amphitheatre Parkway
Mountain View, CA 94043

650-253-0000 main
google.com

that the “clinician unassisted” data in either the search or AMIE condition receive notably different distributions of judgments on each of the three scores, despite the fact that the first task in each condition is that unassisted judgment. I would expect similar results, at least if the subject populations were homogeneous. These results are, nevertheless, roughly in line with the top-n quantitative results.

Response: We appreciate this important consideration regarding parity in baselines for both conditions. While we included breakdowns for Quality, Comprehensiveness and Appropriateness in Figure 4 of the original submission we did not include them for the Top-N accuracy plots. For completeness, we have added dis-aggregated results from the two arms of the study to the Top-N accuracy plots (see Fig. R3 below). Indeed, baseline results for the two experiment arms were obtained from distinct sets of clinicians. This study design decision was intentional to avoid potential learning or other carry-over effects between experimental conditions. We added the following clarification to Section 5.1 Experimental Design explaining measures we took to reduce variability between conditions:

“For the assignment, we formed ten pairs of two clinicians each, grouping clinicians with similar years of post-residency experience together. The set of all cases was then randomly split into ten partitions, and each clinician pair was assigned to one of the ten case partitions. Within each partition, each case was completed once in Condition I by one of the two clinicians, and once in Condition II by the other clinician. The assignment of clinician to one of the two experimental conditions for each case was randomized. Pairing clinicians with similar post-residency experience to complete the same case served to reduce variability between the two distinct experimental conditions.”

Nonetheless, the reviewer correctly notes that numerical results were slightly different between the two baselines which followed identical procedures with distinct sets of clinicians. This is true for both DDX accuracy and subjective rating scores.

(a) The percentage of DDX lists with the final diagnosis through human evaluation.

(b) The percentage of DDX lists with the final diagnosis through automated evaluation.

Figure R3. Top-n Accuracy. Results with unassisted baseline DDX lists from the clinicians in Condition I (Search) and Condition II (AMIE) are broken out.

You are correct that in the two conditions the average scores for the DDX lists were slightly different; however, performing Wilcoxon signed-rank tests the appropriateness scores were statistically similar:

Appropriateness:

Search Baseline = 3.71

AMIE Baseline = 3.75

Wilcoxon signed-rank test $p = 0.63$

The comprehensiveness scores were statically similar:

Comprehensiveness:

The number of cases that scored 4 (i.e., The DDX contains all candidates that are reasonable) was not statistically higher for clinicians in the baseline Search (condition I) and AMIE (II) conditions.

McNemar's Test: $p = 0.23$

And the quality scores were slightly different:

Google LLC
1600 Amphitheatre Parkway
Mountain View, CA 94043

650-253-0000 main
google.com

Quality Score:

The number of cases that scored 5 (i.e., The DDx included the top diagnosis) was higher in the baseline Search (33.8%) compared to the baseline AMIE (27.2%) conditions.

McNemar's Test: $p = 0.03$

These results, along with trends from Figure R3, suggest that baseline distributions were statistically similar with respect to several endpoints, and that for those endpoints where differences could be determined, clinicians in the AMIE condition started off from a slightly *lower* baseline. The latter observation, if anything, may further corroborate (rather than undermine) our main conclusion that AMIE had a stronger assistive effect than Search alone because results after assistance from AMIE were greater than those after assistance with search.

To further consolidate this point we ran a linear mixed effects models to test the effect of the Arm (either Assisted by Search=0 or Assisted by AMIE=1) on the final diagnosis score after assistance while controlling for the effect of baseline (unassisted final diagnosis score).

Assisted Final Diagnosis Score \sim Arm + Unassisted Final Diagnosis Score + (1|Clinician) + ϵ

The results were as follows:

No. Observations: 604, Converged: Yes

	Coefficient	Std. Err.	z	P> z	CI: 0.025	CI 0.975
Intercept	1.438	0.120	12.028	0.000	1.203	1.672
Arm	0.378	0.085	4.451	0.000	0.212	0.545
Baseline	0.620	0.030	20.338	0.000	0.560	0.680
Group Var	0.001					

The Arm has a significant effect on the final diagnosis.
We have added these results to Appendix B.

Comment: *I was confused by the description of their automated evaluation, which generally assigned slightly lower top-n scores to AMIE and slightly higher scores for each condition that includes the judgment of a clinician. From the brief description, it appears that Med-PaLM 2 was*

Google LLC
1600 Amphitheatre Parkway
Mountain View, CA 94043

650-253-0000 main
google.com

used to determine the correctness of each of the up to 10 diagnoses generated by AMIE or by the clinicians with various assistance tools. I cannot figure out what the relationship is between Med-PaLM 2 and the LLM included in AMIE. Are these actually the same? If not, what is the difference between them? If they are the same and the LLM has reasonable internal consistency, why would it judge its own top-1 performance to be only ~27% accurate (Fig 5 right)? I am generally concerned that we don't understand why LLMs can do the kinds of reasoning they seem capable of, so insights would be valuable into this apparent disagreement between an LLM and itself (or perhaps a sibling).

Response: Thank you for emphasizing that further elaboration and discussion of our automated evaluation method is warranted. The LLMs in Med-PaLM 2 and AMIE (our work) are different. They have some similarities (base model architecture and some overlapping training data); however, the AMIE model was tuned on more datasets because they were anticipated to help improve performance for the NEJM cases.

You are correct that Med-PaLM 2 was used to determine the correctness of each of the diagnoses. This was done by feeding both a candidate diagnosis (i.e., a single item from a DDx list) and the correct final diagnosis (per NEJM article) into Med-PaLM 2 and prompting it to determine the binary outcome of whether the candidate diagnosis matched the correct final diagnosis. This binary matching task requires an understanding of medical synonyms and acronyms (e.g. to determine that “COPD” matches “chronic obstructive pulmonary disease”), but not a sophisticated reasoning process as it happens in isolation without access to the NEJM article content itself. We believe that Med-PaLM 2, tuned for medical question answering, lends itself for this constrained matching task. Finally, since the tasks of deriving a DDx list and comparing that list to a final diagnosis are not identical and because we required the model to provide DDx lists even if it were “uncertain” mean that even the same model would not necessarily always judge its top answer to be 100% accurate.

We have added statistical tests to more robustly compare the human and automated evaluation. In particular, we computed Cohen's kappa as a measure of agreement between human raters and automated evaluation with respect to the binary indicator of whether a given diagnosis, i.e. an individual item from a proposed DDx list, matched the correct final diagnosis. Cohen's kappa for this matching task was 0.631 indicating “substantial agreement” between human raters and our automated evaluation method per established guidelines from Landis and Koch (1977). This has been added to Section 5.1.

Google LLC
1600 Amphitheatre Parkway
Mountain View, CA 94043

650-253-0000 main
google.com

Comment: *I was happy to see a significant section on contamination analysis, which is an obvious concern in such studies. For example, the NY Times is apparently suing OpenAI for including copyrighted material in its training corpus, using experimental results that demonstrate that prompting GPT with an initial fragment of an article can pretty well reproduce the entire article. Thus, it is natural to want to know the extent to which the DDX that is calculated by AMIE is simply being regurgitated from something its LLM has ingested previously. I am unsure whether the 512-character overlap analysis was done only on the medical QA data used to fine-tune PaLM 2 or if it applied to the vastly larger training data used in creating PaLM 2 as well. If only done on the fine tuning data, then that comparison is not very convincing.*

Response: We are glad that you appreciated the contamination analysis. We would like to confirm that this was performed on all the pretraining data and also the fine-tuning data. We did not use any NEJM cases for fine-tuning AMIE. Hence, in our description we focused the contamination discussion on the pre-training data. To avoid potential reader confusion, we have clarified in the revision that it is BOTH the pretraining and fine-tuning datasets that were considered for contamination. Please see the edits in Section 6.5.

Comment: *The sentence's meaning "We performed an additional overlap analysis on articles less than, or equal to, 512-characters overlap (N = 249)" is unclear to me. Single-character overlap would presumably find overlap for all N = 302 cases, so what degree of overlap yielded 249? I could imagine other methods of searching for overlap that would not demand character-by-character equality. For example, would sentence or paragraph level computed embedding vectors from each source (or fine-tuning) corpus overlaid on similar vectors for cases yield insights into close paraphrases of the CPCs in other data?*

Response: Thank you for highlighting this important point of potential reader confusion. Perhaps a better way of describing it is that in the contamination analysis we excluded articles with greater than 512- characters overlap (N=53). You are correct that all articles would have at least one character overlap and the wording in our original submission could be confusing. We have revised the description in the revision. The embedding overlap is an interesting idea. However, given the tooling available to us, we were constrained to string matching against the pretraining corpus as our best available option. See edits in Section 6.5 for clarifications made to avoid potential reader confusion.

Google LLC
1600 Amphitheatre Parkway
Mountain View, CA 94043

650-253-0000 main
google.com

Comment: *Of minor note, there is a broken sentence in section 6.4. The brief comparison with GPT-4 shows better performance for top-2 or greater, but the GPT-4 results in fact appear (Fig 7) slightly better for top-1 when judged by either Med-PaLM 2 or GPT 4.*

Response: We are very grateful for the comment on the broken sentence, we believe this was referring to Section 6.6 rather than Section 6.4, which we have fixed. We have also added an acknowledgement that for N=1 GPT-4 results are marginally better but not statistically so. See edits in Section 6.6.

Comment: *The discussion helps to add nuance to the report and points out various limitations, including the unusual difficulty of NEJM CPC cases, the very limited investigation of HCI issues in this work, and the pragmatics of where and how such tools could be used to improve clinical practice. I think one of the important products of the reported research is the determination of gold standard diagnosis judgments for the 302 NEJM cases they used. The Data Availability section reasonably argues that they are unable to disseminate the copyrighted text, but to make their study reproducible and to help advance the field, I would strongly recommend that they publish those judgments so everyone's research can benefit.*

Response: We have uploaded the full set of DDx lists from the physicians and the models as supplemental material to the portal. We agree that these will be a valuable resource to other researchers. The NEJM cases are publicly available and can be obtained from the publisher; however, we do not have license to distribute them to third parties ourselves. The DDx lists can be found in the supplementary material submitted alongside this rebuttal in the Nature portal.

Though tangential to this comment, we note that we have also expanded our qualitative analysis of clinician interviews per guidance from Referee #1 to strengthen investigation of HCI issues in our work. We encourage Referee #2 to review these additions to the manuscript in Methods (5.2 Qualitative Interviews), Results (6.7 Qualitative Analysis) and Discussion.

Thank you again for the opportunity to respond to the reviewers' thoughtful comments. We sought to answer each one in depth. We believe these comments and our responses to them have further strengthened the manuscript, and look forward to hearing from you in due course.

Yours sincerely,

Dr. Daniel McDuff

Google LLC
1600 Amphitheatre Parkway
Mountain View, CA 94043

650-253-0000 main
google.com

Dr. Mike Schaekermann
Dr. Jake Sunshine
Dr. Alan Karthikesalingam
Dr. Vivek Natarajan

Corresponding Authors
Google

Google LLC
1600 Amphitheatre Parkway
Mountain View, CA 94043

650-253-0000 main
google.com

[TEXT REDACTED]

Google LLC
1600 Amphitheatre Parkway
Mountain View, CA 94043

650-253-0000 main
google.com

[TEXT REDACTED]

Google LLC
1600 Amphitheatre Parkway
Mountain View, CA 94043

650-253-0000 main
google.com

[TEXT REDACTED]

Google LLC
1600 Amphitheatre Parkway
Mountain View, CA 94043

650-253-0000 main
google.com

Jan 14th 2025

Dear Editors and Reviewers,

We would like to thank you for your response to our revisions. We are glad you appreciated the analysis in our manuscript and our responses to your comments. We continue to believe that this work is an important step toward validating the efficacy of large language models (LLMs) in supporting differential diagnosis, a foundational task within clinical medicine. We have uploaded our files formatted per the Nature instructions. Below we have responded to your additional comments.

Response to Referee #1

Comment: *In the article, it is mentioned a few times that this model outperforms GPT-4, also referred to as a state-of-the-art model. This may or may not be valid. It is based on the results by Kanjee et al, who used GPT-4 to solve cases. However, I would argue that as of November 2024, this conclusion that it outperforms "GPT-4" is no longer valid based alone on that it performed better than the results found by Kanjee. That is, it lacks support beyond just out-performing the results by Kanjee et al. The reason for this is that GPT-4 has been updated many times since Kanjee et al made their article. I have checked the article, and they unfortunately do not mention which model they used, however they do mention that the model has a learning cutoff of September 2021 which combined with the publication date, leads me to believe that it is GPT-4-0413 - Either way, it is not the current model and even for OpenAI internally it would not be considered state-of-the-art anymore. I think the easiest way to alleviate this is simply to mention clarify this point in the article. Other options would be removing the comparison or actually testing it against the current model. I think this is an important point as OpenAI obviously represents Googles competition.*

Response: The GPT-4 results we used were taken directly from Kanjee et al. [5]. We have clarified in the revision that that paper was published in 2023 and the results are no longer state of the art. The authors did not report the exact version of GPT used, but we have clarified the origin of the results and the date of publication (June 15, 2023) of the article.

Comment: *In the supplementary data, some cases do not have any data for the AMIE or do not contain 10 diagnoses even though that is the prompted answer. I assume this is a simple oversight. (i.e. the lines are just empty). Examples include Case 12-2023 and 29-2015*

Response: Thank you for highlighting this. We have added the missing diagnoses to the supplement file and uploaded the new version.

Google LLC
1600 Amphitheatre Parkway
Mountain View, CA 94043

650-253-0000 main
google.com

Response to Referee #2

Comment: *Figure 4 uses colors to differentiate responses to the rating questions, but gray scale in the x-axis labels. It would help to use the same colors there as well.*

Response: We thank the reviewer for raising this point of potential reader confusion. The colors in Figure 4 correspond to experiment arms (same color coding used throughout the paper) whereas the shade of the color corresponds to different levels on the rating scales. The legend maps the same shades to rating scale labels. It uses gray scale because the rating scales were the same for all experiment arms. We have clarified this in the caption.

Comment: *When describing the different LLMs used in the study (Med-PALM 2, AMIE), it would help the reader to mention some statistics for each, such as the total number of parameters of the model, or the total computational effort including the size of the pre-training data and fine-tuning data or perhaps FLOPS used in training them.*

Response: The base LLM was PaLM 2 [20]. Med-PaLM 2 and AMIE builds upon PaLM 2 [4], a then new iteration of Google's large language model with substantial performance improvements on multiple LLM benchmark tasks. The PaLM series of models have since then been superseded by Gemini including our work on Med-Gemini.

Comment: *There are minor bugs in the references. E.g., the journal is missing in 34 (BMJ, I think), "Hello AI" has non-curly quotes and an extra space in 35, "Anthony Celi, L." should be "Celi, L. A." in 15. I didn't check all of these carefully, but someone should.*

Response: Thanks for pointing out these formatting issues. We have fixed the references.

Yours sincerely,
Dr. Daniel McDuff
Dr. Mike Schaeckermann
Dr. Jake Sunshine
Dr. Alan Karthikesalingam
Dr. Vivek Natarajan

Corresponding Authors
Google